# Flexible time-restricted eating combined with exercise in a free-living setting for middle-aged women with overweight/ obesity: a randomized controlled trial

Zihan Dai [1], Masashi Miyashita[1,2,3], Eric Tsz-chun Poon[1], Xiao Yu Tian [4], Angus Pak-hung Yu[1], Cindy Hui-ping Sit[1] & Stephen Heung-sang Wong [1] ✉

Obesity poses a significant public health challenge among middle-aged women, driven by physiological changes associated with aging and menopause. This parallel-group, assessor-blinded, four-arm randomized controlled trial investigated the effects of 12-week 8-hour flexible time-restricted eating (flexTRE) and aerobic exercise (EX), alone or in combination (flexTRE+EX), on body composition and metabolic health in a free-living setting. Participant enrolment began on September 1st 2023 and data collection was completed on July 1st 2024. Conducted at a single research site in Hong Kong, the trial enrolled women aged 40–60 years with overweight/obesity. Participants were randomized in a 1:1:1:1 ratio to a flexTRE, EX, flexTRE+EX, or control (CON) group (n = 26 per group), with all 104 participants included in the final intention-to-treat analysis. Outcomes were assessed at baseline, and week 12. The primary outcome was fat mass. The flexTRE+EX group achieved the greatest fat mass reduction compared to the CON group (adjusted mean difference [99% Confidence Interval] −2.85 kg [−4.01 to −1.69]), and additional benefit over the flexTRE group alone (−1.56 kg [−2.74 to −0.38]), and the EX group alone (−2.01 kg [−3.21 to −0.81]). Secondary outcomes were reported in the main text. No serious adverse events were reported, and adherence was high (83%-87%) across intervention groups. These findings suggest that the combined approach effectively reduces fat mass and enhances related metabolic parameters, providing a feasible and effective strategy in middle-aged women facing overweight/obesity. Trial registration: ChiCTR2300074846.

Obesity is a complex condition that poses a significant global public health concern. It is associated with an increased risk of chronic diseases, including type 2 diabetes, cardiovascular diseases, certain cancers, and adverse effects on bone health and quality of life[1]. Among various demographic groups, women in midlife are particularly vulnerable to the detrimental effects of overweight or obesity due to

physiological changes associated with aging and menopause transition[2]. On average, these women gain approximately 0.7 kg (1.5 pounds) per year, independent of age at baseline or menopause status, increasing their risk of transitioning from normal or overweight to obesity[3]. While both men and women tend to gain weight in midlife due to aging[4], women experienced additional, unique changes in body

composition driven by hormonal shifts associated with menopause. These shifts result in significantly influence body fat distribution, leading to increased central adiposity[5,6]. As aging and menopause coincide, which further alters body composition in midlife women[5]. This shift is linked to adverse metabolic consequences, including dysglycemia, dyslipidemia, hypertension, and an elevated risk of cardiovascular disease[7].

Given that cardiovascular disease remains the leading cause of death among women[8], the menopausal transition period typically beginning in women's 40 s and progressing from pre- through peri- to post-menopause, represents a critical window for intervention, as it is characterized by a significant increase in cardiovascular risk[9,10]. Effective weight management strategies and early preventive interventions are crucial due to the unfavorable alterations in body composition and metabolic profile in midlife women[11]. For the prevention of weight gain, lifestyle interventions incorporating both dietary modifications and exercise training are key strategies[12]. The optimal diet for weight loss remains debated, as no single approach, whether low-fat, low-carbohydrate, or high-protein has proven superior for sustained results, with differences between diets being marginal[13,14]. Successful weight loss typically depends on adherence to a calorie-restricted plan, regardless of macronutrient composition[12]. Individual preferences and adherence capabilities vary, making optimizing adherence crucial for weight loss success. This is often enhanced by regular professional contact and supportive behavioral change programs[13]. While meal replacements can aid in portion control, they may not be suitable and accessible for everyone in real-life settings. Challenges arise when individuals lack access to professional guidance or tools for calculating macronutrient distribution and calories. Thus, ongoing research is essential to develop sustainable, easy-to-follow strategies for long-term weight management that are accessible to all in real-world scenarios.

Intermittent fasting has gained significant popularity in recent years, with three primary approaches commonly used in human research: alternate-day fasting (ADF), 5:2 fasting, and time-restricted eating (TRE)[15]. While ADF and 5:2 fasting methods typically involve calorie restriction on fasting days and unrestricted eating on non-fasting days, TRE employs a different strategy that emphasizes regulating meal timing rather than calorie counting has become increasingly prevalent[16]. TRE involves restricting the daily eating window to a specified number of hours and fasting with zero-calorie beverages for the remaining hours of the day[17]. While TRE doesn't limit energy intake or macronutrient content, it often leads to a 20–30% reduction in energy consumption, resulting in a 1–4% weight loss over one to 12 weeks[15]. TRE is effective for improving body composition and cardiometabolic health in those with overweight or obesity[18,19], mainly due to energy deficits and aligning eating patterns with circadian rhythms[20]. Unlike conventional calorie-restriction diets necessitating meticulous food tracking, TRE's inherent structure enhances adherence and long-term viability. A previous review also found that TRE showed higher compliance than the 5:2 diet and ADF[21].

Nutrition and exercise are primary therapies for treating overweight and obesity. Aerobic exercise plays a crucial role by inducing physiological adaptations that aid in weight management and reduce body fat through enhanced oxygen utilization[22]. It improves muscle capillarity density, increases mitochondria function, and enhances glucose metabolism and lipid oxidation[23]. Epidemiological evidence demonstrates that women who maintain or increase physical activity during midlife had a lower tendency to gain weight[2,3]. Current guidelines recommend at least 150 min of moderate intensity aerobic exercise, such as brisk walking, weekly for weight loss[24]. While physical activity is essential for weight management, relying solely on exercise without dietary control may not yield significant initial weight loss[25] Similarly, a focus on diet alone is often inadequate without incorporating regular physical activity[2]. An integrated approach is necessary, as

sustained dietary control such as caloric restriction without exercise can reduce basal metabolic rate[26]. This is particularly relevant for aging populations, where declining energy expenditure necessitates concurrent dietary and activity modifications. Given the rising prevalence of obesity among middle-aged women, developing feasible, real-world interventions that are easy to follow and promote good adherence, while enhancing both metabolic health and body composition remains a critical public health priority.

Systematic reviews and meta-analyses have demonstrated that combining dietary and exercise interventions enhances weight loss[27] and improves glycemic outcomes[28] in individuals with obesity. TRE combined with endurance training has shown promising results in trained individuals and athletes, leading to reduced fat mass without compromising performance[29,30], as evidenced by meta-analysis[31]. Studies exploring TRE combined with various exercise modalities, including high-intensity interval training (HIIT)[32,33] and walking[34], have shown potential benefits in individuals with overweight/obesity. Specifically, the combination of HIIT and TRE improved glycemic control and cardiorespiratory fitness[32], while meta-analyses indicate that TRE with exercise positively impacts body composition and metabolic health[35]. Although these studies vary in protocol duration, timing windows, and exercise types across different populations, they consistently suggest synergistic benefits when combining exercise with dietary interventions. However, evidence regarding the metabolic benefits of combined TRE and aerobic exercise interventions specifically in middle-aged women with overweight or obesity, particularly under free-living conditions, remains limited. In addition, previous studies have enforced a fixed eating window (e.g., 12 p.m. to 8 p.m.)[19,36], which may limit the real-world applicability of their findings. To address this, the present study implements a flexible TRE (flexTRE) protocol, allowing participants to self-adjust the timing of their 8 h eating window daily to accommodate the variability of everyday life. Understanding findings under these real-world conditions is crucial as it would be more applicable and readily translatable to practical recommendations for this population. Therefore, this study aims to investigate whether flexTRE and aerobic exercise (EX), alone or in combination (flexTRE+EX), improve body composition and metabolic health outcomes relative to passive control in this specific population in a free-living setting. Specifically, we seek to evaluate whether the combined intervention (flexTRE+EX) produces superior outcomes compared to individual interventions (flexTRE or EX). Through these comparisons, we aim to elucidate the individual and synergistic impacts of these interventions on health outcomes, providing valuable insights for effective lifestyle modification strategies.

Here, we show that combining flexTRE with aerobic exercise is more effective than either intervention alone for reducing adipose tissue in middle-aged women with overweight or obesity. In addition to the primary benefit, the combined intervention led to further improvements in body composition and insulin resistance. This work establishes the combined approach as a feasible, safe, and effective strategy for enhancing metabolic health in a real-world, free-living setting.

## Results
### Participants
A total of 104 eligible volunteers were enrolled in the trial, randomized at a 1:1:1:1 ratio into the flexTRE, EX, flexTRE+EX, and control (CON) groups, each comprising 26 individuals. Ninety-eight participants (94.2%) completed the full 12-week trial duration. Attrition rates were as follows: two participants in flexTRE, two in EX, one in flexTRE+EX, and one in CON did not complete the trial. Participant flow throughout the trial is shown in Fig. 1. Basic characteristics about age, height, body mass, body mass index (BMI), body fat percentage, fat mass, fat free mass, waist circumference, moderate-to-vigorous physical activity per week (MVPA), eating window, daily energy intake, and macronutrient

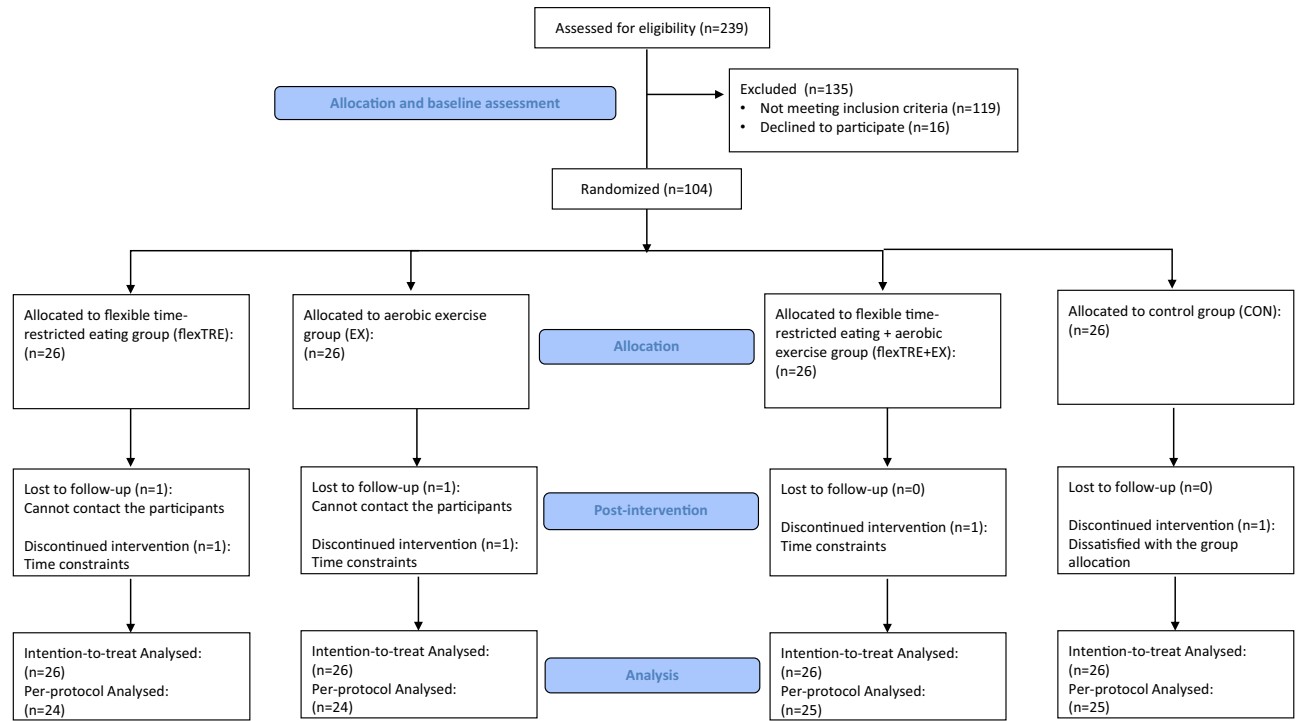

**Fig. 1 | Schematic presentation of the participants screening, randomization, and interventions.**

## Table 1 | Baseline characteristics of participants (*N* = 104)

| Characteristic | flexTRE (*n* = 26) | EX (*n* = 26) | flexTRE + EX (*n* = 26) | CON (*n* = 26) |
|---|---|---|---|---|
| Age, years | 46 ± 5 | 49 ± 6 | 48 ± 6 | 47 ± 6 |
| Height, cm | 159.3 ± 5.8 | 158.5 ± 6.1 | 160.6 ± 5.4 | 158.9 ± 4.8 |
| Body mass, kg | 69.7 ± 7.6 | 67.2 ± 7.9 | 68.3 ± 8.9 | 69.2 ± 11.0 |
| Body mass index, kg/m² | 27.6 ± 3.6 | 26.9 ± 2.6 | 26.3 ± 2.7 | 27.2 ± 3.5 |
| Body fat percentage, % | 38.5 ± 5.5 | 38.3 ± 3.7 | 37.0 ± 3.8 | 38.8 ± 5.4 |
| Fat mass, kg | 27.2 ± 6.7 | 26.0 ± 5.2 | 25.5 ± 6.1 | 27.4 ± 9.1 |
| Fat free mass, kg | 42.6 ± 2.8 | 41.3 ± 3.4 | 42.7 ± 3.3 | 41.9 ± 3.2 |
| Waist circumference, cm | 89.5 ± 6.7 | 87.9 ± 5.5 | 87.6 ± 8.3 | 90.4 ± 9.2 |
| MVPA, min/week | 61.4 ± 58.9 | 78.0 ± 62.9 | 61.7 ± 60.8 | 57.5 ± 72.6 |
| Eating window, h/day | 10.4 ± 1.1 | 10.5 ± 1.4 | 10.6 ± 1.7 | 10.9 ± 1.5 |
| Energy intake, kJ/day | 7173.6 ± 587.6 | 7133.1 ± 452.6 | 7100.3 ± 387.2 | 6907.4 ± 749.1 |
| Carbohydrate percentage | 0.45 ± 0.06 | 0.47 ± 0.08 | 0.44 ± 0.06 | 0.42 ± 0.07 |
| Protein percentage | 0.18 ± 0.04 | 0.17 ± 0.03 | 0.17 ± 0.03 | 0.19 ± 0.04 |
| Fat percentage | 0.37 ± 0.05 | 0.36 ± 0.07 | 0.39 ± 0.06 | 0.39 ± 0.05 |
| Menopausal status, *n* (%) | | | | |
| Premenopausal | 16 (61.5%) | 10 (38.5%) | 17 (65.4%) | 15 (57.7%) |
| Perimenopausal | 6 (23.1%) | 3 (11.5%) | 2 (7.7%) | 4 (15.4%) |
| Postmenopausal | 4 (15.4%) | 13 (50.0%) | 7 (26.9%) | 7 (26.9%) |
| Family history of cardiovascular disease, *n* (%) | | | | |
| With family history | 17 (65.4%) | 19 (73.1%) | 16 (61.5%) | 16 (61.5%) |
| No family history | 9 (34.6%) | 7 (26.9%) | 10 (38.5%) | 10 (38.5%) |

Continuous variables are presented as mean ± standard deviation, while categorical variables are expressed as participants' number (*n*) and percentage (%) in this group.
*CON* control group, *EX* aerobic exercise group, *flexTRE* flexible time-restricted eating group, *flexTRE+EX* flexible time-restricted eating combined with aerobic exercise group.

distribution (carbohydrate, protein, and fat percentage), menopause status, and family history of cardiovascular disease are detailed in Table 1, no significant differences among groups were observed.

**Intention-to-treat analyses.** All randomized participants were included in the intention-to-treat analyses, regardless of completeness of outcome measures.

**Fat mass.** The primary outcome of this study is fat mass. A group-by-time interaction effect was observed in fat mass (*P* < 0.001). Compared with the control group, all intervention groups including flexTRE, EX, and flexTRE+EX showed a larger reduction in fat mass (flexTRE vs. control: adjusted mean difference [99% confidence interval], −1.29 kg [−2.33 to −0.25]; *P* = 0.001; EX vs. control: −0.84 kg [−1.89 to 0.20]; *P* = 0.038; flexTRE+EX vs. control: −2.85 kg [−4.01 to −1.69]; *P* < 0.001).

In addition, the combined group showed a larger reduction in fat mass than the flexTRE and EX group (flexTRE+EX vs. flexTRE: −1.56 kg [−2.74 to −0.38]; $P < 0.001$; flexTRE+EX vs. EX: −2.01 kg [−3.21 to −0.81]; $P < 0.001$) (Table 2). The percentage changes in fat mass for each participant across the groups over the 12-week intervention are illustrated in Fig. 2. Notably, the flexTRE+EX group achieved a mean reduction in fat mass of 2.7 kg, representing a 10.2% decrease.

## Other body composition parameters

Group-by-time interaction effects were also observed in other body composition parameters, including body mass ($P < 0.001$), BMI ($P < 0.001$), body fat percentage ($P < 0.001$), fat free mass ($P = 0.013$), waist circumference ($P < 0.001$), and hip circumference ($P < 0.001$). At post-intervention, all these outcomes, except for fat free mass, showed significantly greater reductions in the three intervention groups (flexTRE, EX, flexTRE+EX) compared to the control group. The combined group showed greater reductions in body mass, BMI, body fat percentage, and waist circumference than either the flexTRE or EX groups individually. Details results are presented in Table 2. Regarding fat-free mass (FFM), only the flexTRE+EX group showed a larger reduction compared to the CON group (−0.49 kg [−0.89 to −0.09]; $P = 0.002$). Figure 2 illustrates the percentage changes in body mass for each participant across the groups over the 12-week intervention. Notably, the minimal clinically important difference (MCID) for body weight is typically defined as a reduction of over 5% (shown as red line)[37]. In the flexTRE+EX group, the mean weight loss was 3 kg, representing a 4.4% reduction, which did not meet the MCID threshold.

## Glycemic control

No significant group-by-time interaction effect was observed in the fasting glucose and hemoglobin A1c (HbA1c) levels ($P = 0.453$ and 0.865, respectively). Regarding insulin, group-by-time interaction effect was observed ($P = 0.023$), the combined group (flexTRE+EX) exhibited a larger reduction compared to the other two intervention groups (flexTRE+EX vs. flexTRE: −2.98 mU/L [−5.06 to −0.89]; $P < 0.001$; flexTRE+EX vs. EX: −2.84 mU/L [−5.25 to −0.42]; $P = 0.003$) at post-intervention (Table 3). Similarly, group-by-time interaction effects were also observed in the homeostatic model assessment of insulin resistance (HOMA-IR) and Quantitative insulin sensitivity check index (QUICKI) ($P = 0.029$ and $P = 0.012$, respectively). At the post-intervention, the combined group showed a larger reduction compared to the other two intervention groups (flexTRE+EX vs. flexTRE: −0.72 [−1.26 to −0.18]; $P < 0.001$; flexTRE+EX vs. EX: −0.75 [−1.35 to −0.15]; $P = 0.001$) in HOMA-IR and showed a larger increase compared to the two intervention groups (flexTRE+EX vs. flexTRE: 0.03 [0.01 to 0.05]; $P < 0.001$; flexTRE+EX vs. EX: 0.02 [0.003 to 0.04]; $P = 0.003$) in QUICKI (Table 3).

## Cardiometabolic parameters

There were no significant group-by-time interaction effects in resting heart rate, systolic blood pressure, or diastolic blood pressure ($P = 0.323$, $P = 0.252$, and 0.799, respectively). Lipid profiles, including total cholesterol (TC), triglycerides (TG), high-density lipoprotein cholesterol (HDL), and low-density lipoprotein cholesterol (LDL), were meticulously examined in this study, with results detailed in Table 4. Notably, no significant group-by-time interaction effects were noted in TC, HDL, LDL, or TG ($P = 0.863$, $P = 0.785$, $P = 0.082$, and 0.495, respectively).

## Adipokines

A group-by-time interaction effect was found in leptin ($P < 0.001$). Leptin concentration showed a significantly large reduction in all intervention groups including flexTRE, EX, and flexTRE+EX when compared with the control group (flexTRE vs. control: −4.73 ng/ml [−9.26 to −0.19]; $P = 0.007$; EX vs. control: −4.89 ng/ml [−9.23 to −0.54]; $P = 0.004$; flexTRE+EX vs. control: −8.22 ng/ml [−11.88 to −4.56]; $P < 0.001$) at post-intervention. Additionally, the combined group showed a larger reduction in leptin concentration compared to the flexTRE group (flexTRE+EX vs. flexTRE: −3.49 ng/ml [−7.18 to 0.19]; $P = 0.015$) and EX group (flexTRE+EX vs. EX: −3.33 ng/ml [−6.55 to −0.12]; $P = 0.008$). Regarding adiponectin levels, no significant group-by-time interaction effect was found ($P = 0.134$). Although we found significant group-by-time interaction effect in resistin ($P = 0.026$), however after the Holm procedure to adjust multiple comparisons, there were no between group differences (Table 5).

## Sleep quality, quality of life, and mood profile

To provide insights into the broader effects of the intervention on overall well-being, several validated instruments were used to assess participants' sleep, quality of life, and mood profile (Table 6). The Pittsburgh Sleep Quality Index (PSQI)[38] was used to evaluate the sleep quality of the participants. However, no significant group-by-time interaction effect was found ($P = 0.225$). The World Health Organization Quality of Life Brief Version (WHOQOL-BREF)[39] was used to assess the quality of life of the participants; still, no group-by-time interaction effect was found ($P = 0.077$). In addition, there was no significant group-by-time interaction effect found in the mood profile ($P = 0.349$).

## Energy intake

A group-by-time interaction effect was observed in energy intake ($P < 0.001$). The flexTRE group and combined group (flexTRE+EX) exhibited a larger reduction compared to the control group (flexTRE vs. CON: −801.39 kJ [−1296.37 to −306.41]; $P < 0.001$; flexTRE+EX vs. CON: −998.63 kJ [−1506.95 to −490.31]; $P < 0.001$) at post-intervention (Table 7). The combined group also produced greater reductions in energy intake than the EX group (flexTRE+EX vs. EX: −719.47 kJ [−1269.39 to −169.55]; $P < 0.001$).

## Macronutrient composition

No significant group-by-time interaction effects were found in the macronutrient composition, including carbohydrates, protein, and fat percentage ($P = 0.151$; $P = 0.162$; $P = 0.123$, respectively) (Table 7).

## Eating window

A group-by-time interaction effect was observed in eating window ($P < 0.001$). The flexTRE and flexTRE+EX groups exhibited significantly greater reductions in eating windows compared to the EX and CON groups. Details could be found in Table 7.

## Moderate-to-vigorous physical activity

A group-by-time interaction effect was observed in MVPA level as well ($P < 0.001$). Specifically, the EX and flexTRE+EX groups exhibited significantly greater increases in MVPA level compared to the flexTRE and CON groups (Table 7).

## Subgroup analysis

We conducted a subgroup analysis based on the participants' menopausal status. When a significant interaction effect between menopausal status-by-group-by-time was identified, we further analyzed this outcome within subgroups. Detailed results of this analysis are provided in Supplementary Table S1.

## Per-protocol analyses

The secondary per-protocol (PP) analyses included only those participants who completed all baseline and post-intervention assessments. Ninety-eight participants (flexTRE, $n = 24$; EX, $n = 24$; flexTRE+EX, $n = 25$; CON, $n = 25$) were included in the PP analyses (Fig. 1). Results from the PP analyses were no different from the ITT analyses (Supplementary Table S2).

**Table 2 | Data related to fat mass and other body composition parameters**

| Measurement | Group | Baseline n | Baseline Mean | Baseline SD | Postintervention n | Postintervention Mean | Postintervention SD | Time effect | Group effect | Time × group effect | Compared with CON Adjusted mean difference | Compared with CON 99%CI | Compared with CON P value | Compared with flexTRE + EX Adjusted mean difference | Compared with flexTRE + EX 99%CI | Compared with flexTRE + EX P value |
|---|---|---|---|---|---|---|---|---|---|---|---|---|---|---|---|---|
| **Primary outcome** | | | | | | | | | | | | | | | | |
| Fat mass | flexTRE | 26 | 27.24 | 6.71 | 24 | 25.46 | 6.73 | *P* < 0.001 | *P* < 0.001 | *P* < 0.001 | -1.29 | -2.33 to -0.25 | **0.001*** | 1.56 | 0.38 to 2.74 | **<0.001*** |
| kg | EX | 26 | 25.96 | 5.24 | 24 | 25.16 | 5.55 | | | | -0.84 | -1.89 to 0.20 | **0.038*** | 2.01 | 0.81 to 3.21 | **<0.001*** |
| | flexTRE+EX | 26 | 25.53 | 6.11 | 25 | 22.92 | 5.30 | | | | -2.85 | -4.01 to -1.69 | **<0.001*** | | | |
| | CON | 26 | 27.40 | 9.06 | 25 | 27.70 | 9.48 | | | | | | | | | |
| **Secondary outcomes** | | | | | | | | | | | | | | | | |
| Body composition Body mass | flexTRE | 26 | 69.66 | 7.62 | 24 | 67.63 | 7.77 | *P* < 0.001 | *P* < 0.001 | *P* < 0.001 | -1.59 | -2.51 to -0.66 | **<0.001*** | 1.78 | 0.64 to 2.92 | **<0.001*** |
| kg | EX | 26 | 67.24 | 7.86 | 24 | 66.49 | 8.28 | | | | -1.27 | -2.19 to -0.36 | **<0.001*** | 2.09 | 0.92 to 3.27 | **<0.001*** |
| | flexTRE+EX | 26 | 68.26 | 8.93 | 25 | 65.21 | 8.35 | | | | -3.37 | -4.50 to -2.23 | **<0.001*** | | | |
| | CON | 26 | 69.24 | 10.98 | 25 | 69.83 | 11.32 | | | | | | | | | |
| Body mass index | flexTRE | 26 | 27.63 | 3.62 | 24 | 26.75 | 3.65 | *P* < 0.001 | *P* < 0.001 | *P* < 0.001 | -0.79 | -1.11 to -0.47 | **<0.001*** | 0.64 | 0.14 to 1.13 | **<0.001*** |
| kg/m² | EX | 26 | 26.87 | 2.56 | 24 | 26.54 | 2.69 | | | | -0.5 | -0.84 to -0.15 | **<0.001*** | 0.93 | 0.42 to 1.44 | **<0.001*** |
| | flexTRE+EX | 26 | 26.28 | 2.73 | 25 | 25.07 | 2.50 | | | | -1.43 | -1.91 to -0.95 | **<0.001*** | | | |
| | CON | 26 | 27.21 | 3.55 | 25 | 27.48 | 3.62 | | | | | | | | | |
| Body fat percentage | flexTRE | 26 | 38.49 | 5.55 | 24 | 37.15 | 5.69 | *P* < 0.001 | *P* < 0.001 | *P* < 0.001 | -1.14 | -2.10 to -0.18 | **0.002*** | 1.26 | 0.03 to 2.49 | **0.008*** |
| % | EX | 26 | 38.33 | 3.68 | 24 | 37.48 | 4.04 | | | | -0.75 | -1.45 to -0.05 | **0.006*** | 1.65 | 0.69 to 2.62 | **<0.001*** |
| | flexTRE+EX | 26 | 36.98 | 3.82 | 25 | 34.80 | 3.61 | | | | -2.4 | -3.36 to -1.45 | **<0.001*** | | | |
| | CON | 26 | 38.85 | 5.43 | 25 | 38.94 | 5.59 | | | | | | | | | |
| Fat free mass | flexTRE | 26 | 42.65 | 2.82 | 24 | 42.16 | 2.98 | *P* = 0.031 | *P* = 0.002 | *P* = 0.013 | -0.43 | -0.99 to 0.13 | 0.048 | 0.06 | -0.58 to 0.70 | 0.804 |
| kg | EX | 26 | 41.28 | 3.43 | 24 | 41.33 | 3.56 | | | | -0.25 | -0.57 to 0.06 | 0.036 | 0.24 | -0.17 to 0.64 | 0.130 |
| | flexTRE+EX | 26 | 42.71 | 3.35 | 25 | 42.29 | 3.52 | | | | -0.49 | -0.89 to -0.09 | **0.002*** | | | |
| | CON | 26 | 41.87 | 3.20 | 25 | 42.12 | 3.14 | | | | | | | | | |
| Waist circumference | flexTRE | 26 | 89.53 | 6.71 | 24 | 84.63 | 6.80 | *P* < 0.001 | *P* < 0.001 | *P* < 0.001 | -3.24 | -6.07 to -0.40 | **0.003*** | 2.94 | -0.01 to 5.88 | **0.010*** |
| cm | EX | 26 | 87.90 | 5.49 | 24 | 83.75 | 5.65 | | | | -3.21 | -5.64 to -0.78 | **<0.001*** | 2.97 | 0.43 to 5.51 | **0.003*** |
| | flexTRE+EX | 26 | 87.60 | 8.29 | 25 | 80.24 | 8.07 | | | | -6.18 | -8.81 to -3.54 | **<0.001*** | | | |
| | CON | 26 | 90.40 | 9.19 | 25 | 89.04 | 8.07 | | | | | | | | | |
| Hip circumference | flexTRE | 26 | 106.81 | 7.28 | 24 | 101.92 | 6.01 | *P* < 0.001 | *P* < 0.001 | *P* < 0.001 | -3.26 | -5.57 to -0.95 | **<0.001*** | 1.96 | -0.44 to 4.37 | 0.036 |
| cm | EX | 26 | 104.50 | 4.76 | 24 | 99.83 | 4.52 | | | | -3.66 | -5.84 to -1.49 | **<0.001*** | 1.56 | -0.76 to 3.87 | 0.083 |
| | flexTRE+EX | 26 | 104.43 | 6.14 | 25 | 98.08 | 5.35 | | | | -5.22 | -7.76 to -2.69 | **<0.001*** | | | |
| | CON | 26 | 106.35 | 9.28 | 25 | 105.12 | 8.99 | | | | | | | | | |

Intervention effects on these outcomes were examined by generalized estimating equations (GEE). The model assessed the main effects of group and time, as well as the group-by-time interaction, while adjusting for covariates including baseline outcome values, age, menopausal status, and family history of cardiovascular disease. A significant group-by-time interaction indicated a significant difference for a given outcome between interventions over time. Pairwise treatment comparisons were performed by linear contrasts using the Holm procedure to adjust for multiple comparisons, and raw *P* values are presented. Bold *P*-values indicate a significant main effect of time, group, or a group-by-time interaction (*P* < 0.05). An asterisk (*) in addition to a bold *P*-value indicates a significant pairwise difference between intervention groups over time after applying the Holm procedure for multiple comparisons.
*CI* confidence interval, *CON* control group, *EX* aerobic exercise group, *flexTRE* flexible time-restricted eating group, *flexTRE+EX* flexible time-restricted eating combined with aerobic exercise group, *SD* standard deviation.

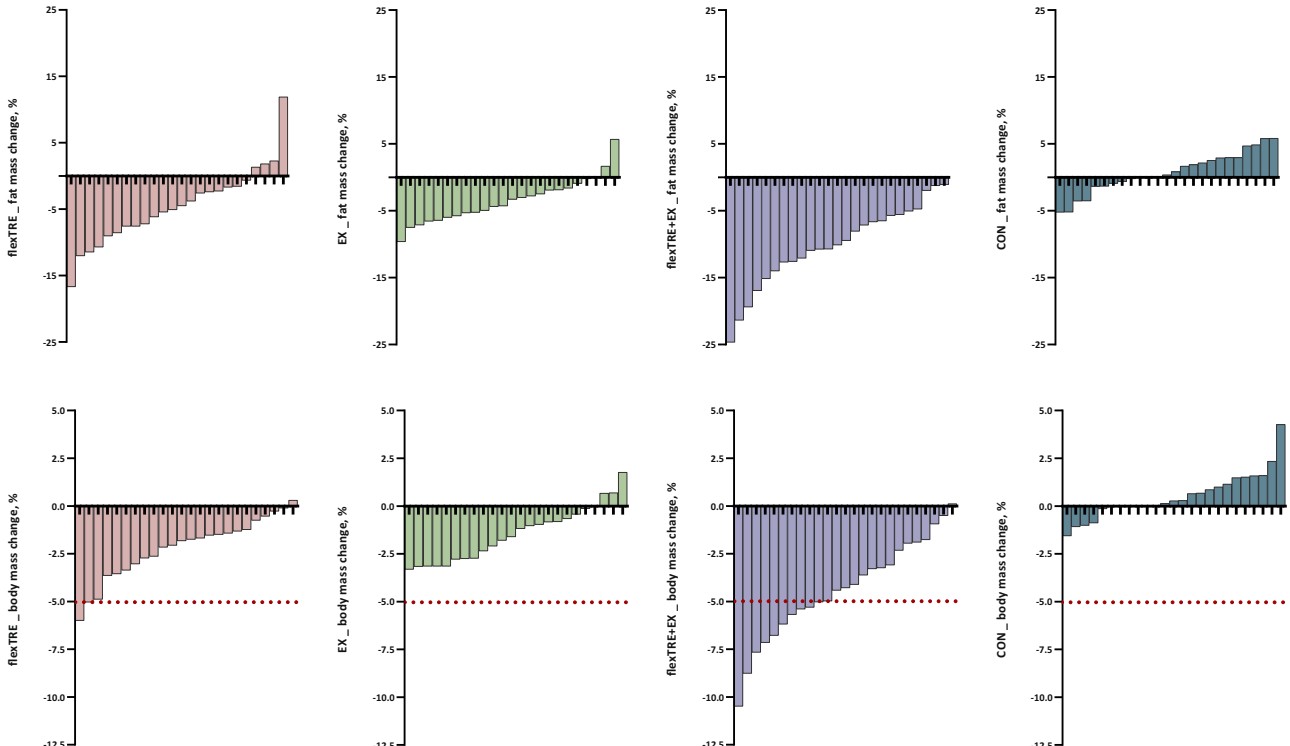

**Fig. 2 | Percentage changes in fat mass and body mass for each participant.**
Note: CON control group, EX aerobic exercise group, flexTRE flexible time-restricted eating group, flexTRE+EX flexible time-restricted eating combined with aerobic exercise group. The upper four charts display the percentage changes in fat mass for individual participants in the flexTRE, EX, flexTRE+EX, and CON groups after the 12-week intervention. Each data point represents the raw fat mass change for a single participant. The lower four charts illustrate the percentage changes in body mass for the same groups over the 12-week intervention. A red line is included in the lower charts to indicate a 5% body mass reduction, which represents the threshold for a minimally clinically important difference. The raw data could be found in source data file.

## Adverse events

No serious adverse events were reported throughout the study duration. Minor adverse events included heightened feelings of hunger that might have impacted sleep in six participants undergoing the flexTRE and flexTRE+EX interventions during the initial two weeks. Additionally, two individuals in the EX group reported slight discomfort and tiredness during the first two weeks of exercise sessions, expressing difficulty in completing the entire session. These minor adverse events were primarily observed during the early stages of the intervention and subsided as participants adapted to the new regimens.

## Adherence rate

Adherence to the intervention protocols was assessed for each group over the 12-week period. In the flexTRE group, adherence was calculated based on participants' daily eating windows. On average, participants in the flexTRE group achieved an average adherence rate of 86.9%. For the EX group, participants completed an average of 83.7% of the prescribed sessions (36 sessions over 12 weeks). In the combination group (flexTRE+EX), adherence was assessed separately for flexTRE and EX components. Participants in the flexTRE+EX group achieved an average adherence rate of 89.3% for flexTRE and 79.6% for EX, resulting in an overall average adherence rate of 84.6% for this group. These results indicate high adherence rates across all groups, demonstrating the feasibility of implementing these lifestyle interventions over a 12-week period. In addition, participants in the EX group and flexTRE+EX group engaged in aerobic exercise as prescribed, with flexibility to choose from brisk walking, jogging, or running, ensuring the prescribed heart rate was achieved. In the EX group (n = 24), 6 participants (25.0%) chose brisk walking, 11 participants (45.8%) chose jogging, and 4 participants (16.7%) chose running. Two participants (8.3%) opted to use an elliptical machine rather than running-related activities, as they reported discomfort or difficulty within the first two weeks. Additionally, one participant (4.2%) didn't do any exercise as prescribed during the intervention period. In the flexTRE+EX group (n = 25), 1 participant (4.0%) performed brisk walking, 19 participants (76.0%) performed jogging, and 5 participants (20.0%) performed running.

## Discussion

The present study evaluated and compared the independent and combined impacts of flexTRE and EX on body composition and metabolic health among middle-aged women with overweight and obesity over a 12-week period in a free-living setting. Significantly, the combined flexTRE+EX intervention produced superior outcomes in reducing fat mass and improving other body composition parameters, as well as metabolic markers (such as decreased insulin, HOMA-IR, leptin) compared to either intervention alone. Analysis also revealed significant spontaneous reductions in total energy intake in both flexTRE and flexTRE+EX groups, with macronutrient composition remaining consistent. High protocol adherence (>83% across all groups. 86.9%, 84.6%, and 83.7% for flexTRE+EX, flexTRE, and EX groups respectively) and absence of severe adverse events highlight the feasibility, safety, and effectiveness of this combined approach in real-world settings. These findings suggest that the flexTRE+EX intervention offers a promising strategy for addressing obesity-related health concerns.

The combined flexTRE+EX intervention demonstrated superior fat mass loss (−10.2%) compared to individual interventions (flexTRE: −4.6%, EX: −3.5%). Similarly, total body mass reduction was also greater in the flexTRE+EX group (−4.4%) versus flexTRE (−2.2%) and EX (−1.5%)

**Table 3 | Data related to glycemic control**

| Measurement | Group | Baseline | | | Postintervention | | | Time effect | Group effect | Time × group effect | Postintervention | | | | | |
| | | | | | | | | | | | Compared with CON | | | Compared with flexTRE + EX | | |
| | | n | Mean | SD | n | Mean | SD | | | | Adjusted mean difference | 99%CI | P value | Adjusted mean difference | 99%CI | P value |
| Fasting glucose mmol/L | flexTRE | 26 | 5.50 | 0.78 | 24 | 5.43 | 0.72 | P = 0.469 | P = 0.333 | P = 0.453 | 0.05 | −0.18 to 0.28 | 0.583 | 0.09 | −0.18 to 0.36 | 0.397 |
| | EX | 26 | 5.36 | 0.41 | 24 | 5.45 | 0.44 | | | | 0.14 | −0.12 to 0.40 | 0.170 | 0.18 | −0.11 to 0.46 | 0.111 |
| | flexTRE+EX | 26 | 5.35 | 0.55 | 25 | 5.21 | 0.48 | | | | −0.04 | −0.31 to 0.23 | 0.703 | | | |
| | CON | 26 | 5.21 | 0.47 | 25 | 5.18 | 0.50 | | | | | | | | | |
| Insulin mIU/L | flexTRE | 26 | 8.79 | 5.29 | 24 | 9.77 | 5.12 | P = 0.112 | **P = 0.003** | **P = 0.023** | 1.42 | −0.85 to 3.70 | 0.107 | 2.98 | 0.89 to 5.06 | **<0.001\*** |
| | EX | 26 | 8.54 | 4.90 | 24 | 9.14 | 3.56 | | | | 1.28 | −1.27 to 3.83 | 0.195 | 2.84 | 0.42 to 5.25 | **0.003\*** |
| | flexTRE+EX | 26 | 8.40 | 6.11 | 25 | 6.52 | 3.64 | | | | −1.55 | −3.69 to 0.59 | 0.061 | | | |
| | CON | 26 | 8.47 | 5.13 | 25 | 8.54 | 5.22 | | | | | | | | | |
| HOMA-IR | flexTRE | 26 | 2.21 | 1.53 | 24 | 2.46 | 1.59 | p = 0.152 | **P = 0.004** | **p = 0.029** | 0.34 | −0.24 to 0.93 | 0.130 | 0.72 | 0.18 to 1.26 | **<0.001\*** |
| | EX | 26 | 2.02 | 1.13 | 24 | 2.21 | 0.90 | | | | 0.37 | −0.26 to 1.01 | 0.132 | 0.75 | 0.15 to 1.35 | **0.001\*** |
| | flexTRE+EX | 26 | 2.10 | 1.82 | 25 | 1.52 | 0.89 | | | | −0.38 | −0.92 to 0.17 | 0.074 | | | |
| | CON | 26 | 2.01 | 1.34 | 25 | 2.03 | 1.38 | | | | | | | | | |
| QUICKI | flexTRE | 26 | 0.36 | 0.04 | 24 | 0.35 | 0.03 | P = 0.906 | **P = 0.005** | **P = 0.012** | −0.01 | −0.03 to 0.01 | 0.160 | −0.03 | −0.05 to −0.01 | **<0.001\*** |
| | EX | 26 | 0.35 | 0.03 | 24 | 0.35 | 0.02 | | | | −0.01 | −0.02 to 0.01 | 0.352 | −0.02 | −0.04 to −0.003 | **0.003\*** |
| | flexTRE+EX | 26 | 0.36 | 0.05 | 25 | 0.38 | 0.05 | | | | 0.02 | −0.01 to 0.04 | 0.063 | | | |
| | CON | 26 | 0.36 | 0.05 | 25 | 0.36 | 0.04 | | | | | | | | | |
| HbA1c mmol/mol | flexTRE | 26 | 36.77 | 4.79 | 24 | 36.08 | 5.56 | P = 0.017 | P = 0.576 | P = 0.865 | 0.68 | −1.24 to 2.59 | 0.361 | 0.12 | −1.87 to 2.12 | 0.872 |
| | EX | 26 | 37.00 | 3.38 | 24 | 36.29 | 3.59 | | | | 0.30 | −1.72 to 2.32 | 0.703 | −0.25 | −2.32 to 1.81 | 0.750 |
| | flexTRE+EX | 26 | 36.31 | 4.63 | 25 | 35.72 | 3.85 | | | | 0.55 | −0.99 to 2.10 | 0.356 | | | |
| | CON | 26 | 35.35 | 3.60 | 25 | 34.32 | 3.64 | | | | | | | | | |

Intervention effects on these outcomes were examined by generalized estimating equations (GEE). The model assessed the main effects of group and time, as well as the group-by-time interaction, while adjusting for covariates including baseline outcome values, age, menopausal status, and family history of cardiovascular disease. A significant group-by-time interaction indicated a significant difference for a given outcome between interventions over time. Pairwise treatment comparisons were performed by linear contrasts using the Holm procedure to adjust for multiple comparisons, and raw P values are presented. Bold P-values indicate a significant main effect of time, group, or a group-by-time interaction (P < 0.05). An asterisk (*) in addition to a bold P-value indicates a significant pairwise difference between intervention groups over time after applying the Holm procedure for multiple comparisons.

*CI* confidence interval, *CON* control group, *EX* aerobic exercise group, *flexTRE* flexible time-restricted eating group, *flexTRE+EX* flexible time-restricted eating combined with aerobic exercise group, *HbA1c* Hemoglobin A1c, *HOMA-IR* Homeostatic Model Assessment of Insulin Resistance, *QUICKI* Quantitative insulin sensitivity check index, *SD* standard deviation.

**Table 4 | Data related to cardiometabolic parameters**

| Measurement | Group | Baseline | | | Postintervention | | | Time effect | Group effect | Time × group effect | Postintervention Compared with CON | | | Compared with flexTRE + EX | | |
|---|---|---|---|---|---|---|---|---|---|---|---|---|---|---|---|---|
| | | n | Mean | SD | n | Mean | SD | | | | Adjusted mean difference | 99%CI | P value | Adjusted mean difference | 99%CI | P value |
| Systolic BP mmHg | flexTRE | 26 | 107.00 | 14.03 | 24 | 101.08 | 18.60 | P < 0.001 | P = 0.169 | P = 0.252 | −5.69 | −14.73 to 3.34 | 0.104 | 1.02 | −6.71 to 8.74 | 0.734 |
| | EX | 26 | 110.19 | 15.91 | 24 | 104.33 | 16.75 | | | | −5.50 | −15.41 to 4.41 | 0.153 | 1.21 | −6.78 to 9.21 | 0.696 |
| | flexTRE+EX | 26 | 110.38 | 12.93 | 25 | 102.28 | 13.56 | | | | −6.71 | −14.96 to 1.53 | 0.036 | | | |
| | CON | 26 | 109.08 | 16.44 | 25 | 106.48 | 12.25 | | | | | | | | | |
| Diastolic BP mmHg | flexTRE | 26 | 76.46 | 8.77 | 24 | 76.17 | 10.67 | P = 0.432 | P = 0.697 | P = 0.799 | −0.79 | −8.27 to 6.68 | 0.784 | 2.02 | −5.21 to 9.25 | 0.473 |
| | EX | 26 | 75.38 | 10.42 | 24 | 75.42 | 13.60 | | | | −1.42 | −9.10 to 6.27 | 0.634 | 1.39 | −5.77 to 8.56 | 0.617 |
| | flexTRE+EX | 26 | 76.62 | 7.30 | 25 | 74.48 | 11.37 | | | | −2.81 | −9.90 to 4.28 | 0.307 | | | |
| | CON | 26 | 78.35 | 11.57 | 25 | 77.88 | 13.22 | | | | | | | | | |
| Resting heart rate rpm | flexTRE | 26 | 72.50 | 8.97 | 24 | 69.50 | 6.14 | P = 0.081 | P = 0.237 | P = 0.323 | −2.43 | −6.51 to 1.65 | 0.125 | −0.68 | −5.39 to 4.03 | 0.709 |
| | EX | 26 | 70.46 | 8.76 | 24 | 70.88 | 8.62 | | | | 0.71 | −3.51 to 4.93 | 0.664 | 2.46 | −2.33 to 7.24 | 0.186 |
| | flexTRE+EX | 26 | 72.19 | 8.85 | 25 | 70.36 | 7.98 | | | | −1.75 | −5.88 to 2.39 | 0.277 | | | |
| | CON | 26 | 72.00 | 7.80 | 25 | 71.76 | 8.78 | | | | | | | | | |
| Total cholesterol mmol/L | flexTRE | 26 | 5.83 | 1.21 | 24 | 5.66 | 0.77 | P = 0.346 | P = 0.859 | P = 0.863 | −0.05 | −0.47 to 0.37 | 0.747 | 0.02 | −0.40 to 0.43 | 0.923 |
| | EX | 26 | 5.86 | 0.78 | 24 | 5.91 | 0.88 | | | | 0.04 | −0.34 to 0.42 | 0.769 | 0.11 | −0.26 to 0.48 | 0.440 |
| | flexTRE+EX | 26 | 5.52 | 0.82 | 25 | 5.42 | 0.89 | | | | −0.07 | −0.48 to 0.35 | 0.672 | | | |
| | CON | 26 | 5.57 | 0.94 | 25 | 5.55 | 1.13 | | | | | | | | | |
| HDL cholesterol mmol/L | flexTRE | 26 | 1.44 | 0.35 | 24 | 1.41 | 0.29 | P = 0.019 | P = 0.993 | P = 0.785 | 0.02 | −0.08 to 0.13 | 0.564 | 0.02 | −0.09 to 0.13 | 0.620 |
| | EX | 26 | 1.47 | 0.32 | 24 | 1.44 | 0.35 | | | | 0.01 | −0.11 to 0.13 | 0.823 | 0.01 | −0.12 to 0.14 | 0.881 |
| | flexTRE+EX | 26 | 1.56 | 0.36 | 25 | 1.52 | 0.29 | | | | 0.003 | −0.12 to 0.12 | 0.947 | | | |
| | CON | 26 | 1.59 | 0.32 | 25 | 1.51 | 0.31 | | | | | | | | | |
| LDL cholesterol mmol/L | flexTRE | 26 | 3.64 | 0.99 | 24 | 3.55 | 0.80 | P = 0.166 | P = 0.090 | P = 0.082 | −0.15 | −0.53 to 0.24 | 0.325 | 0.16 | −0.23 to 0.54 | 0.296 |
| | EX | 26 | 3.69 | 0.63 | 24 | 3.62 | 0.71 | | | | −0.18 | −0.49 to 0.12 | 0.125 | 0.12 | −0.17 to 0.42 | 0.292 |
| | flexTRE+EX | 26 | 3.26 | 0.68 | 25 | 3.04 | 0.67 | | | | −0.30 | −0.61 to −0.001 | 0.010 | | | |
| | CON | 26 | 3.33 | 0.68 | 25 | 3.44 | 0.87 | | | | | | | | | |
| Triglycerides mmol/L | flexTRE | 26 | 1.11 | 0.52 | 24 | 1.13 | 0.37 | P = 0.472 | P = 0.040 | P = 0.495 | 0.15 | −0.11 to 0.40 | 0.136 | 0.20 | −0.04 to 0.45 | 0.033 |
| | EX | 26 | 1.36 | 0.50 | 24 | 1.28 | 0.52 | | | | 0.04 | −0.16 to 0.24 | 0.698 | 0.10 | −0.17 to 0.37 | 0.352 |
| | flexTRE+EX | 26 | 1.28 | 0.98 | 25 | 1.21 | 0.70 | | | | −0.06 | −0.29 to 0.18 | 0.529 | | | |
| | CON | 26 | 1.21 | 0.62 | 25 | 1.16 | 0.49 | | | | | | | | | |

Intervention effects on these outcomes were examined by generalized estimating equations (GEE). The model assessed the main effects of group and time, as well as the group-by-time interaction, while adjusting for covariates including baseline outcome values, age, menopausal status, and family history of cardiovascular disease. A significant group-by-time interaction indicated a significant difference for a given outcome between interventions over time. Pairwise treatment comparisons were performed by linear contrasts using the Holm procedure to adjust for multiple comparisons, and raw P values are presented. Bold P-values indicate a significant main effect of time, group, or a group-by-time interaction (P < 0.05). An asterisk (*) in addition to a bold P-value indicates a significant pairwise difference between intervention groups over time after applying the Holm procedure for multiple comparisons. CI confidence interval, CON control group, EX aerobic exercise group, HDL high-density lipoprotein, flexTRE flexible time-restricted eating group, flexTRE+EX flexible time-restricted eating combined with aerobic exercise group, LDL low-density lipoprotein, SD standard deviation.

groups. In the flexTRE+EX group, a body mass loss of 4.4% (3 kg) and a fat mass loss of 10.2% (2.7 kg) suggest that the intervention is particularly effective at promoting fat loss relative to overall weight loss. Although the observed weight loss did not meet the minimal clinically important difference (MCID), the findings remain promising, especially as they are based on all participants in a free-living setting without considering adherence rates. Our findings align with previous research that reported a 4.58% weight loss when combining ADF with aerobic exercise[40]. However, our study demonstrated superior outcomes with flexTRE+EX compared to both individual interventions, whereas the previous study showed enhanced effects only versus exercise alone. While ADF typically associated with greater weight loss than TRE[21], the superior results of our combined intervention suggest the effectiveness of pairing TRE with exercise to optimize weight and fat loss outcomes. The reduction in waist circumference, an indirect estimate of visceral fat, was notable across all interventions: flexTRE +EX (7.5 cm), flexTRE (4.8 cm), and EX (4.6 cm) groups. Notably, the flexTRE+EX group achieved an 8.4% reduction in waist circumference following the 12-week intervention. This result is particularly significant, as previous research has shown that a 10% increase in waist circumference is associated with a 1.48 (95% CI 1.36–1.61) times higher mortality risk in middle-aged women and men[41]. Furthermore, a reduction in waist circumference is a critically important treatment target for reducing adverse health risks[42], particularly for women undergoing the menopausal transition, where increased visceral fat poses a significant health concern.

Interestingly, our subgroup analysis revealed that EX alone did not significantly improve body composition in post-menopausal participants compared to controls. This aligns with findings from recent research by Jozwiak et al., which reported that exercise alone failed to improve body composition in menopausal women; however, when combined with TRE, the benefits were more pronounced[43]. Previous studies have suggested that hormonal changes during menopause may reduce the positive adaptations to physical exercise, potentially limiting its efficacy in post-menopausal women[44]. In contrast, TRE alone or combined with exercise remained effective, emphasizing the critical role of dietary interventions in managing body composition in this population. TRE may modulate metabolic pathways and improve metabolic flexibility, complementing exercise adaptations. These findings suggest that post-menopausal women may benefit more from dietary-based strategies, such as TRE, combined with exercise interventions, rather than relying exclusively on exercise to improve body composition and metabolic health.

We observed a modest (<0.5 kg) but statistically significant reduction in FFM across all three intervention groups before correcting for multiple comparisons. This reduction was proportional to total weight loss, which is consistent with the established physiological observation that ~25% of lost weight is derived from FFM[45], suggesting the finding may not be clinically meaningful. After applying the Holm procedure for multiple comparisons, only the combined flexTRE+EX group retained a statistically significant reduction in FFM. The loss of significance in the flexTRE and EX groups alone likely reflects limited statistical power to detect small effects in individual interventions. Future randomized controlled trials with larger sample sizes and longer follow-up durations are warranted to further evaluate the complex interplay between these interventions and FFM.

Although none of the interventions (flexTRE, EX, flexTRE+EX) showed significantly larger improvement in insulin sensitivity compared to controls, likely due to relatively normal baseline glycemic profiles, the combined intervention provided additional benefits compared to flexTRE or EX alone, characterized by lower fasting insulin concentrations and HOMA-IR scores, along with increased QUICKI. The enhanced insulin sensitivity observed in the combined intervention group can be attributed to complementary physiological mechanisms of both interventions. Aerobic exercise improves insulin

sensitivity through multiple pathways, including enhanced expression and activity of glucose-regulating proteins in skeletal muscle, improved muscular fat oxidation capacity, and increased capillary density that facilitates glucose diffusion from capillaries to muscle cells[46]. While previous research on isolated TRE interventions in adults with obesity showed minimal changes in fasting glucose levels over 2-12 months, with significant improvements in insulin parameters primarily observed only in protocols implementing shorter eating windows scheduled earlier in the day[16]. TRE reduces insulin concentrations by moderating the frequency of insulin spikes, thereby enhancing insulin sensitivity[47]. Consistent adherence to TRE over the long term may help sustain lower and more stable insulin concentrations, consequently reducing the risk of insulin resistance and type 2 diabetes[47]. Additionally, TRE aligns food intake with periods of heightened insulin sensitivity, thereby improving glucose tolerance[48]. Therefore, the combined approach likely further optimizes metabolic function even in relatively healthy populations. Interestingly, subgroup analysis revealed unexpected results in peri-menopausal participants, potentially due to limited sample size and increased variability. Larger studies are needed to confirm this subgroup-specific phenomenon and clarify potential hormonal influences in peri-menopausal women. However, we observed no significant differences in fasting glucose concentrations or HbA1c levels, suggesting that the interventions primarily influenced insulin dynamics rather than long-term glycemic control. Moreover, this finding may be attributed to the participants' normoglycemic status at baseline (fasting glucose less than 5.6 mmol/L, HbA1c less than 42 mmol/mol (6.5%), which limited the potential for substantial changes in glycemic outcomes. For individuals at risk of metabolic disorders, comprehensive lifestyle modifications might be beneficial for promoting metabolic enhancements, although the effects may be less pronounced in a healthier population compared to those with existing metabolic conditions. Future studies should also employ continuous glucose monitoring to assess mean glucose levels, time within prespecified glucose ranges, and multiple measures of glycemic variability to draw more solid conclusions[49].

Prior research shows TRE has minimal impact on plasma lipids, with some studies reporting unchanged HDL and LDL levels[18,36,50], while others noted increased LDL[51,52]. In our study, we found no significant between-group effects for HDL and LDL, though the flexTRE +EX group likely lower LDL post-intervention. This finding aligns with the hypothesis that combining TRE with exercise may enhance lipid metabolism through improved lipoprotein lipase activity[53]. Although exercise is typically associated with increased HDL levels in individuals with obesity, no significant changes were observed in our study, possibly due to optimal baseline levels or the moderate intensity of the exercise intervention[54,55]. Analysis of adipokines unveiled noteworthy reductions in leptin concentrations across the flexTRE, EX, and flexTRE +EX groups compared to the control group, with the flexTRE+EX group showing the greatest decrease. Leptin, a hormone that regulates appetite and energy expenditure via the hypothalamus, typically declines with weight loss[56]. Evidence suggests that both TRE and exercise independently contribute to reductions in leptin concentrations[57,58], likely reflecting improvements in body composition and energy balance. In addition to leptin, adipokines such as resistin, which is linked to inflammation and insulin resistance, also likely to decrease with improvements in adiposity and insulin sensitivity[59]. These changes in adipokine profiles highlight broader metabolic benefits, including reduced inflammation and improved metabolic regulation[60]. The greater impact observed in the flexTRE+EX group highlights the potential anti-inflammatory and metabolic advantages of combining TRE with exercise. In this population of middle-aged women experiencing aging and menopause, attention to both metabolic health and psychological well-being is important. Although not statistically significant, the interventions appeared to

**Table 5 | Data related to adipokines**

| Measurement | Group | Baseline | | | Postintervention | | | Time effect | Group effect | Time × group effect | Postintervention Compared with CON | | | Compared with flexTRE + EX | | |
|---|---|---|---|---|---|---|---|---|---|---|---|---|---|---|---|---|
| | | n | Mean | SD | n | Mean | SD | P value | P value | P value | Adjusted mean difference | 99%CI | P value | Adjusted mean difference | 99%CI | P value |
| Adiponectin ug/ml | flexTRE | 24 | 4.97 | 3.03 | 24 | 5.21 | 2.36 | P < 0.001 | P = 0.391 | P = 0.134 | −0.27 | −1.47 to 0.93 | 0.564 | −0.76 | −2.34 to 0.83 | 0.219 |
| | EX | 24 | 5.50 | 2.42 | 24 | 5.46 | 1.82 | | | | −0.35 | −1.21 to 0.51 | 0.299 | −0.83 | −2.20 to 0.53 | 0.116 |
| | flexTRE+EX | 25 | 5.89 | 4.27 | 25 | 6.7 | 3.82 | | | | 0.49 | −0.95 to 1.92 | 0.381 | | | |
| | CON | 25 | 5.93 | 2.24 | 25 | 6.36 | 2.83 | | | | | | | | | |
| Leptin ng/ml | flexTRE | 24 | 19.24 | 12.85 | 24 | 16.49 | 13.21 | P < 0.001 | P < 0.001 | P < 0.001 | −4.73 | −9.26 to −0.19 | 0.007* | 3.49 | 0.19 to 7.18 | 0.015* |
| | EX | 24 | 19.86 | 10.29 | 24 | 14.43 | 5.64 | | | | −4.89 | −9.23 to −0.54 | 0.004* | 3.33 | 0.12 to 6.55 | 0.008* |
| | flexTRE+EX | 25 | 16.21 | 5.26 | 25 | 9.99 | 5.10 | | | | −8.22 | −11.88 to −4.56 | <0.001* | | | |
| | CON | 25 | 21.02 | 13.00 | 25 | 22.45 | 14.33 | | | | | | | | | |
| Resistin ng/ml | flexTRE | 24 | 4.63 | 2.91 | 24 | 5.82 | 4.14 | P = 0.134 | P = 0.433 | P = 0.026 | 1.03 | −0.77 to 2.83 | 0.142 | 1.80 | −0.09 to 3.69 | 0.014 |
| | EX | 24 | 3.82 | 2.18 | 24 | 3.60 | 2.03 | | | | 0.11 | −1.68 to 1.90 | 0.877 | 0.88 | −0.97 to 2.72 | 0.220 |
| | flexTRE+EX | 25 | 5.84 | 5.39 | 25 | 4.69 | 4.00 | | | | −0.77 | −2.48 to 0.94 | 0.245 | | | |
| | CON | 25 | 6.62 | 6.23 | 25 | 6.94 | 6.30 | | | | | | | | | |

Intervention effects on these outcomes were examined by generalized estimating equations (GEE). The model assessed the main effects of group and time, as well as the group-by-time interaction, while adjusting for covariates including baseline outcome values, age, menopausal status, and family history of cardiovascular disease. A significant group-by-time interaction indicated a significant difference for a given outcome between interventions over time. Pairwise treatment comparisons were performed by linear contrasts using the Holm procedure to adjust for multiple comparisons, and raw P-values are presented. Bold P-values indicate a significant main effect of time, group, or group-by-time interaction (P < 0.05). An asterisk (*) in addition to a bold P-value indicates a significant pairwise difference between intervention groups over time after applying the Holm procedure for multiple comparisons.
CI confidence interval, CON control group, EX aerobic exercise group, flexTRE flexible time-restricted eating group, flexTRE+EX flexible time-restricted eating combined with aerobic exercise group, SD standard deviation.

improve quality of life, particularly in peri-menopausal participants. While these were not primary outcomes, future studies should explore these aspects further.

The key strengths of this study include its free-living design, allowing participants to self-select their 8-hour eating window (before 8 PM) with ad libitum meal consumption, which enhanced real-world applicability. Daily communication with participants facilitated adherence to the intervention and monitored the control group properly. Additionally, the use of wearable technology enabled effective monitoring of exercise compliance and provided feedback on weekly adherence assessments. However, several limitations warrant consideration. First, the inclusion of participants at various menopausal status (pre, peri-, post-) may introduce physiological variability. Although subgroup analyses were conducted, the imbalance and small sample size among groups limit the robustness of these conclusions. Nevertheless, this approach provides a comprehensive insight into females in their mid-life with overweight/obesity. Second, variations in TRE eating window timing may have influenced outcomes, necessitating future secondary analyses to examine this factor. Lastly, the study's generalizability is limited to middle-aged East Asian women. Ethnic differences, for example in insulin secretion and sensitivity[61], even within Asian populations, should be considered when interpreting these results.

In summary, combining flexible time-restricted eating with aerobic exercise effectively enhances body composition by reducing fat mass and improves related metabolic parameters in middle-aged women with overweight and obesity. The combined intervention demonstrated high adherence and absence of severe adverse events in real-world settings, suggesting it's high feasibility and practicality. These effects highlight the potential of this approach for promoting comprehensive lifestyle changes to combat obesity and related metabolic conditions. This practical strategy offers healthcare systems a promising alternative for preventing and managing chronic diseases in this demographic.

## Methods
### Study design
This single-center, parallel-group, assessor-blinded, four-arm randomized controlled trial was conducted at a single research site in Hong Kong in accordance with the detailed study protocol (Supplementary Note 1) to investigate and compare the isolated and combined effects of 12 weeks of flexTRE and EX in a free-living setting on body composition and metabolic health in middle-aged women with overweight and obesity. The study followed the principles of the Declaration of Helsinki. Ethical approval was obtained from the Joint Chinese University of Hong Kong-New Territories East Cluster Clinical Research Ethics Committee (reference number 2023.238-T), and the study was registered at the Chinese Clinical Trial Registry (ChiCTR2300074846). Participant enrolment began on September 1st 2023 and data collection was completed on July 1st 2024. The reporting of this trial follows the Consolidated Standards of Reporting Trials (CONSORT) 2010 guidelines, and the completed checklist can be found in Supplementary Note 2.

### Participant recruitment and randomization
Participants were recruited from the general community through various means, including advertisement via posters on campus and social media. Preliminary eligibility screening was conducted using an online questionnaire that assessed physical activity levels, health status, and obesity status. Individuals who passed the online screening were invited to a laboratory visit for further evaluation. Those meeting the inclusion criteria were provided with detailed study information and gave written informed consent voluntarily before enrollment. Eligible participants were randomly assigned to one of four groups in a 1:1:1:1 ratio using an online random number generator with a block size

**Table 6 | Data related to sleep quality, quality of life, and mood profile**

| Measurement | Group | Baseline | | | Postintervention | | | Time effect | Group effect | Time × group effect | Postintervention | | | | | |
|---|---|---|---|---|---|---|---|---|---|---|---|---|---|---|---|---|
| | | | | | | | | | | | Compared with CON | | | Compared with flexTRE + EX | | |
| | | n | Mean | SD | n | Mean | SD | | | | Adjusted mean difference | 99%CI | P value | Adjusted mean difference | 99%CI | P value |
| Sleep quality PSQI | flexTRE | 26 | 6.46 | 2.93 | 23 | 6.48 | 3.31 | P = 0.651 | P = 0.130 | P = 0.225 | 0.02 | −2.03 to 2.14 | 0.948 | −0.91 | −3.33 to 1.51 | 0.333 |
| | EX | 25 | 7.40 | 2.45 | 24 | 6.33 | 2.84 | | | | −0.96 | −2.91 to 0.99 | 0.204 | −1.92 | −4.19 to 0.34 | 0.028 |
| | flexTRE+EX | 26 | 6.62 | 3.46 | 24 | 7.58 | 3.67 | | | | 0.96 | −1.33 to 3.26 | 0.280 | | | |
| | CON | 26 | 7.31 | 3.12 | 25 | 7.12 | 3.59 | | | | | | | | | |
| Quality of life WHOQOL-BREF | flexTRE | 26 | 90.00 | 12.73 | 24 | 90.46 | 13.09 | P = 0.017 | P = 0.122 | P = 0.077 | 1.01 | −3.36 to 5.38 | 0.550 | −1.97 | −7.07 to 3.13 | 0.320 |
| | EX | 26 | 86.54 | 10.05 | 24 | 91.13 | 10.99 | | | | 4.28 | −0.68 to 9.25 | 0.026 | 1.30 | −4.23 to 6.83 | 0.546 |
| | flexTRE+EX | 26 | 89.88 | 11.44 | 24 | 92.08 | 12.60 | | | | 2.98 | −1.34 to 7.31 | 0.075 | | | |
| | CON | 25 | 85.88 | 11.87 | 25 | 85.52 | 12.61 | | | | | | | | | |
| Mood profile POMS | flexTRE | 25 | 125.48 | 17.54 | 24 | 127.83 | 18.24 | P = 0.570 | P = 0.193 | P = 0.349 | 5.63 | −5.20 to 16.46 | 0.181 | −2.06 | −13.70 to 9.57 | 0.648 |
| | EX | 23 | 120.78 | 15.58 | 24 | 121.38 | 13.87 | | | | 1.03 | −8.74 to 10.80 | 0.786 | −6.66 | −17.56 to 4.24 | 0.115 |
| | flexTRE+EX | 26 | 123.35 | 12.16 | 25 | 128.32 | 21.10 | | | | 7.69 | −4.04 to 19.42 | 0.091 | | | |
| | CON | 25 | 125.16 | 16.59 | 25 | 123.44 | 13.69 | | | | | | | | | |

Intervention effects on these outcomes were examined by generalized estimating equations (GEE). The model assessed the main effects of group and time, as well as the group-by-time interaction, while adjusting for covariates including baseline outcome values, age, menopausal status, and family history of cardiovascular disease. A significant group-by-time interaction indicated a significant difference for a given outcome between interventions over time. Pairwise treatment comparisons were performed by linear contrasts using the Holm procedure to adjust for multiple comparisons, and raw P-values are presented. Bold P-values indicate a significant main effect of time, group, or a group-by-time interaction (P < 0.05). An asterisk (*) in addition to a bold P-value indicates a significant pairwise difference between intervention groups over time after applying the Holm procedure for multiple comparisons.
CI confidence interval, CON control group, EX aerobic exercise group, flexTRE flexible time-restricted eating group, flexTRE+EX flexible time-restricted eating combined with aerobic exercise group, POMS Profile of Mood States, PSQI Pittsburgh Sleep Quality Index, SD standard deviation, WHOQOL-BREF World Health Organization Quality of Life Brief Version.

**Table 7 | Data related to additional monitor parameters including energy intake, macronutrient distribution, eating window, and MVPA**

| Measurement | Group | Baseline | | | Postintervention | | | Time effect | Group effect | Time × group effect | Postintervention | | | | | |
| | | | | | | | | | | | Compared with CON | | | Compared with flexTRE + EX | | |
| | | n | Mean | SD | n | Mean | SD | | | | Adjusted mean difference | 99%CI | P value | Adjusted mean difference | 99%CI | P value |
|---|---|---|---|---|---|---|---|---|---|---|---|---|---|---|---|---|
| Energy intake | flexTRE | 26 | 7173.57 | 587.60 | 24 | 6361.06 | 742.20 | P < 0.001 | P < 0.001 | P < 0.001 | −801.39 | −1296.37 to −306.41 | **<0.001*** | 197.24 | −384.43 to 778.91 | 0.382 |
| kJ/day | EX | 26 | 7133.11 | 452.64 | 24 | 6863.54 | 587.50 | | | | −279.16 | −730.24 to 171.92 | 0.111 | 719.47 | 169.55 to 1269.39 | **<0.001*** |
| | flexTRE+EX | 26 | 7100.29 | 387.22 | 25 | 6154.50 | 754.59 | | | | −998.63 | −1506.95 to −490.31 | **<0.001*** | | | |
| | CON | 26 | 6907.37 | 749.13 | 25 | 7024.74 | 626.74 | | | | | | | | | |
| Carbohydrate % | flexTRE | 26 | 0.45 | 0.06 | 24 | 0.42 | 0.07 | P = 0.559 | P = 0.181 | P = 0.151 | −0.045 | −0.10 to 0.01 | 0.050 | −0.042 | −0.10 to 0.02 | 0.086 |
| | EX | 26 | 0.47 | 0.08 | 24 | 0.47 | 0.07 | | | | −0.009 | −0.06 to 0.04 | 0.621 | −0.006 | −0.06 to 0.04 | 0.762 |
| | flexTRE+EX | 26 | 0.44 | 0.06 | 25 | 0.46 | 0.08 | | | | −0.003 | −0.06 to 0.05 | 0.873 | | | |
| | CON | 26 | 0.42 | 0.07 | 25 | 0.45 | 0.07 | | | | | | | | | |
| Protein % | flexTRE | 26 | 0.18 | 0.04 | 24 | 0.19 | 0.04 | P = 0.894 | P = 0.215 | P = 0.162 | 0.017 | −0.01 to 0.05 | 0.114 | 0.002 | −0.03 to 0.03 | 0.883 |
| | EX | 26 | 0.17 | 0.03 | 24 | 0.17 | 0.04 | | | | 0.000 | −0.02 to 0.02 | 0.965 | −0.015 | −0.04 to 0.01 | 0.114 |
| | flexTRE+EX | 26 | 0.17 | 0.03 | 25 | 0.18 | 0.04 | | | | 0.016 | −0.01 to 0.04 | 0.091 | | | |
| | CON | 26 | 0.19 | 0.04 | 25 | 0.18 | 0.04 | | | | | | | | | |
| Fat % | flexTRE | 26 | 0.37 | 0.05 | 24 | 0.39 | 0.06 | P = 0.436 | P = 0.395 | P = 0.123 | 0.029 | −0.02 to 0.08 | 0.123 | 0.038 | −0.01 to 0.09 | 0.056 |
| | EX | 26 | 0.36 | 0.07 | 24 | 0.37 | 0.07 | | | | 0.009 | −0.03 to 0.05 | 0.556 | 0.018 | −0.02 to 0.06 | 0.257 |
| | flexTRE+EX | 26 | 0.39 | 0.06 | 25 | 0.36 | 0.07 | | | | −0.009 | −0.05 to 0.03 | 0.590 | | | |
| | CON | 26 | 0.39 | 0.05 | 25 | 0.37 | 0.06 | | | | | | | | | |
| Eating window | flexTRE | 26 | 10.37 | 1.08 | 24 | 7.88 | 1.47 | P < 0.001 | P < 0.001 | P < 0.001 | −2.02 | −2.97 to −1.05 | **<0.001*** | 0.43 | −0.67 to 1.53 | 0.315 |
| h/day | EX | 26 | 10.52 | 1.40 | 24 | 10.19 | 1.95 | | | | 0.32 | −0.74 to 1.38 | 0.434 | 2.77 | 1.56 to 3.99 | **<0.001*** |
| | flexTRE+EX | 26 | 10.64 | 1.71 | 25 | 7.64 | 1.64 | | | | −2.45 | −3.56 to −1.35 | **<0.001*** | | | |
| | CON | 26 | 10.93 | 1.50 | 25 | 10.32 | 1.41 | | | | | | | | | |
| MVPA | flexTRE | 26 | 61.42 | 58.91 | 23 | 63.16 | 45.05 | P < 0.001 | P < 0.001 | P < 0.001 | 1.44 | −24.89 to 27.76 | 0.888 | −202.14 | −292.78 to −111.51 | **<0.001*** |
| | EX | 26 | 78.05 | 62.92 | 23 | 209.42 | 88.62 | | | | 130.02 | 85.76 to 174.29 | **<0.001*** | −73.56 | −167.60 to 20.49 | 0.044 |
| | flexTRE+EX | 26 | 61.66 | 60.80 | 21 | 211.01 | 112.43 | | | | 203.58 | 113.41 to 293.75 | **<0.001*** | | | |
| | CON | 26 | 57.50 | 72.63 | 24 | 62.44 | 49.18 | | | | | | | | | |

Intervention effects on these outcomes were examined by generalized estimating equations (GEE). The model assessed the main effects of group and time, as well as the group-by-time interaction, while adjusting for covariates including baseline outcome values, age, menopausal status, and family history of cardiovascular disease. A significant group-by-time interaction indicated a significant difference for a given outcome between interventions over time. Pairwise treatment comparisons were performed by linear contrasts using the Holm procedure to adjust for multiple comparisons, and raw P values are presented. Bold P-values indicate a significant main effect of time, group, or a group-by-time interaction (P < 0.05). An asterisk (*) in addition to a bold P-value indicates a significant pairwise difference between intervention groups over time after applying the Holm procedure for multiple comparisons.
CI confidence interval, CON control group, EX aerobic exercise group, flexTRE flexible time-restricted eating group, flexTRE+EX flexible time-restricted eating combined with aerobic exercise group, MVPA moderate-to-vigorous physical activity, SD standard deviation.

of 8, conducted by an independent researcher. Group assignments were disclosed to participants only after they completed all baseline assessments, using sealed envelopes to ensure confidentiality. Outcome assessors were blinded to group allocation to minimize bias.

## Participants

Inclusion criteria: 1) Chinese females aged 40–60 years old; 2) Obesity/overweight: body mass index $\geq$23 kg/m$^2$; 3) physically inactive: $\leq$150 min/week of moderate- to vigorous-intensity physical activity; 4) Weight stable for 3 months prior to the beginning of the study (gain or loss <4 kg); 5) Baseline eating period is $\geq$ 10 h per day.

The exclusion criteria of participants include 1) Known cardiovascular disease; 2) Type 1 or 2 diabetes mellitus; 3) Currently taking medications that could affect study outcomes. For example, anti-hypertension medication and taking glucose-lowering or lipid-lowering medication; 4) Known musculoskeletal disease or injury that will affect the exercise; 5) Night shift workers; 6) Breastfeeding, pregnant or trying to become pregnant; 7) Smokers; 8) Currently on a special or prescribed diet for other reasons.

We amended the inclusion criterion from $\geq$ 12 h to $\geq$ 10 h to better reflect real-life settings in the community. During recruitment for our trial, we found that most participants had eating windows of 10–12 h, with few exceeding 12 h. This adjustment enhances the study's feasibility and maintains its scientific integrity, ensuring our findings are relevant and applicable to common eating practices.

## Sample size estimation

Sample size estimation was conducted using G*Power software based on an effect size derived from a previous study examining the combined effects of ADF with endurance exercise in adults with obesity[62]. Based on these data, an effect size for the group-by-time interaction was calculated (Effect size $f = 0.18$). Using this effect size, a power analysis determined that 23 participants per group would be required to achieve 80% statistical power at a two-sided significance level of $\alpha = 0.05$. To account for an anticipated dropout rate of 10%, the study aimed to enroll a total of at least 102 participants, with 26 participants allocated to each of the four groups. The power analysis was reperformed to generate a more conservative sample size estimation, ensuring an adequate sample size for the predesignated pairwise comparisons.

## Control group/waiting list

Participants assigned to the control group (waitlist control) were instructed to maintain their usual physical activity levels and dietary habits throughout the study. They did not receive any specific dietary or exercise guidance during this period. Weekly check-ins were conducted to confirm continued participation and adherence. Upon completing the study, participants in the waitlist control group were offered the opportunity to receive an appropriate intervention as part of the study's follow-up support.

## flexTRE group

Participants allocated to the flexTRE group were advised to follow ad libitum eating pattern within an 8-hour eating window, employing the 16/8 intermittent fasting regimen. No restrictions on food types or quantities were imposed during the 8-hour eating period, and participants were not required to monitor energy intake. Throughout fasting periods, participants were encouraged to stay hydrated with water and allowed energy-free beverages like black tea and coffee. Participants had the flexibility to choose their preferred 8-hour eating window, starting between 8:00 AM and 12:00 PM and ending between 4:00 PM and 8:00 PM. To better simulate real-world conditions, they were allowed to adjust their eating window daily, provided it remained within the 8:00 AM to 8:00 PM range. Consistency in the exact timing of the eating window was not required. To enhance adherence,

participants were expected to log their daily eating start and stop times online, with researchers monitoring logs daily and sending reminders to non-compliant participants. Participants maintained the flexTRE strategy until the evening before the post-intervention assessments.

## EX group

Participants in the exercise group were asked to engage in aerobic exercise three times a week at a moderate intensity (64–76% of HR$_{max}$) for 40 min per session. Each participant was equipped with a heart rate monitor (Unite, Polar, Finland), with data synchronization allowing researchers to conduct weekly adherence assessments. A progressive exercise approach was implemented, with an initial focus on 30 min walking or running sessions for the first two weeks, progressing to the full exercise program for the subsequent 10 weeks. Each exercise session included a 5 min warm-up and cool-down, aiming to accumulate ~150 min of moderate-intensity exercise per week in alignment with physical activity guidelines[63]. Permitted activities include running, jogging, or brisk walking indoors or outdoors, ensuring the prescribed heart rate is achieved. Dietary interventions were not provided, and participants were asked to maintain their customary dietary patterns. Assessments after the intervention period were undertaken 48-72 hours after the last exercise session for participants in this group.

## flexTRE + EX group

Participants in the combined intervention group simultaneously underwent the prescribed flexTRE regimen and exercise plan. Given the 16 h fasting requirement, participants were advised to schedule exercise sessions between 8 am and 10 pm to mitigate the risk of hypoglycemia. Assessments after the intervention period were undertaken 48-72 h after the last exercise session for participants in this group.

## Outcomes

Pre- and post-intervention evaluations were conducted to assess various outcome measures.

## Primary outcome

Fat mass was evaluated following an 8 h fast, with participants refraining from consuming any liquids, using a bioelectrical impedance analyzer (MC-780 MA, Tanita, Japan) that has been previously validated compared to dual-energy x-ray absorptiometry [fat mass (kg) ICC: 0.925][64]. Measurements were conducted following a standardized protocol at the same time each morning. Participants wore light, form-fitting attire and remained barefoot, with 0.5 kg consistently deducted to account for their clothing. Prior to each measurement, participants' date of birth, gender, and height were entered into the analyzer. The "normal" body type setting (rather than "athlete") and the Asian judgment setting were applied. Body mass index (BMI) was calculated according to the World Health Organization (WHO) guidelines, ensuring a consistency and standardized assessment across all tests.

## Secondary outcomes

Secondary outcomes include the other body composition parameters such as body mass, body fat percentage, fat-free mass, BMI, waist and hip circumferences, waist-hip ratio and outcomes encompassed biomarkers associated with metabolic health, comprising fasting glucose, HbA1c, insulin, HOMA-IR, QUICKI, TC, TG, HDL, LDL, leptin, and adiponectin, and resistin. Blood pressure and resting heart rate measurements were also recorded. Additionally, subjective questionnaires were utilized to evaluate the quality of life, sleep quality, and mood profile.

Other body composition parameters assessment: Height measurements were taken using a stadiometer (Seka, Leicester, UK) to the nearest 0.1 cm, with the participant standing up straight against a wall. Body mass, body fat percentage, and fat-free mass were evaluated

using the same bioelectrical impedance analyzer as fat mass (MC-780 MA, Tanita, Japan). BMI was computed using the formula BMI=weight (kg)/height (m)². Waist and hip circumferences were measured with participants in a standing position with arms at the sides, feet positioned close together, and weight evenly distributed across the feet using a tension tape accurate to 0.1 cm[65]. Waist circumference was measured at the approximate midpoint between the lower margin of the last palpable rib and the top of the iliac crest[65]. Hip circumference was measured at the widest portion of the buttocks posteriorly in a horizontal plane. The waist-hip ratio was calculated accordingly.

Venous blood samples were obtained from an antecubital vein by a registered nurse following an overnight fast of at least 8 h. Participants were instructed to abstain from alcohol, caffeine, and exercise within 24 h prior to blood collection. The blood samples were centrifuged using a Thermo Scientific™ centrifuge (Waltham, USA) at 3500 rpm for 12 min at 4 °C. Plasma was aliquoted into Eppendorf tubes. Freshly prepared plasma was used for measuring fasting glucose and HbA1c. The rest of the plasma samples were stored at −80 °C until the contents were analyzed.

Plasma glucose concentrations and HbA1c levels were promptly measured post-blood sample collection using portable analyzers (Contour Plus Glucometer, Bayer Healthcare, Germany) and the Cobas B 101 HbA1c testing system (Roche Diagnostics, Rotkreuz, Switzerland), respectively. Insulin was measured using commercial ELISA kits (Mercodia, Uppsala, Sweden). HOMA-IR and QUICKI used the formula HOMA-IR= [fasting insulin (µg/ml)]*[fasting glucose (mmol/l)]/22.5[66] and QUICKI = 1/[log(fasting insulin (µU/ml) + log(fasting glucose (mg/dl)][67].

Lipid profiles, including TC, TG, HDL, and LDL, were measured using a Cobas c 111 analyzer (Roche Diagnostics, Rotkreuz, Switzerland). Leptin and adiponectin concentrations were measured using ELISA kits (R&D Systems, Minneapolis, US). Resistin concentration was measured using ELISA kits (BYabscience, Nanjing, China).

Blood pressure and resting heart rate were assessed after a 10-minute rest using a clinical automatic blood pressure monitor (M7 Intelli IT, Omron, Japan). The cuff was placed around the participants' brachial artery (left arm). Two readings (with 1-minute intervals) of systolic blood pressure (SBP), diastolic blood pressure (DBP), and resting heart rate were averaged.

To assess Subjective sleep, quality of life, and mood profile, the Chinese version of the Pittsburgh Sleep Quality Index (PSQI) was used to measure participants' sleep quality[38]. Moreover, WHOQOL-BREF was used to measure participants' Quality of Life[39]. Regarding the mood profile, it was assessed through the modified Profile of Mood States (POMS). The modified POMS is a widely recognized tool designed to evaluate mood states across various dimensions, including tension, depression, anger, vigor, fatigue, confusion, and esteem-related affect[68].

## Monitor parameters
We reclassified energy intake and macronutrient distribution as additional monitor parameters to better reflect their role in providing supplementary insights rather than serving as key endpoints. To enhance measurement feasibility and reflect the free-living nature of our intervention, we also incorporated two new monitor parameters: eating window and moderate-to-vigorous physical activity (MVPA) level. This change does not affect the study's primary or secondary outcome analyses or conclusions.

To assess energy intake, participants were instructed to record their energy intake using a seven-day food record during the week before and the last week of intervention. Each time, participants were asked to record their daily energy intake with ingredients and quantities, including meals, snacks, and drinks, and send pictures of their meals to the investigators. Completed dietary records were analyzed using the online nutrition database of the Centre for Food Safety, Hong Kong (http://www.cfs.gov.hk/). The total energy intake and macronutrient distribution were calculated.

**Eating window.** Participants were instructed to record their daily eating window for one week prior to the intervention (baseline) and throughout the 12-week (84-day) intervention. The daily eating window was defined as the interval between the first and last reported food intake of each day. To assess behavior change, eating window data from the seven days before the intervention and the final 7 days of the intervention were compared. The records from the entire 12-week intervention period also served to evaluate adherence to the prescribed eating window.

**MVPA.** During the week prior to intervention and the final week of the intervention, participants were asked to wear an ActiGraph GT3X-BT accelerometer (ActiGraph Corp, Pensacola, FL, USA) on their non-dominant wrist for seven consecutive days (baseline and post-intervention). Participants removed the device only during water-based activities (e.g., showering, bathing, swimming) and recorded the non-wearing times and reasons in an activity log. The accelerometer recorded raw acceleration along three axes at a sampling frequency of 100 Hz, producing files in.gt3x format. These files were downloaded using ActiLife software (ActiGraph Corp, Pensacola, FL, USA) and subsequently analyzed in R (4.4.1) with the GGIR package[69]. Time spent in MVPA was defined by a mean acceleration of ≥100.6 mg, based on a laboratory calibration study of adults aged 21–61 years[70]. To reduce the likelihood of misclassifying brief artifacts as MVPA, activity was further classified as MVPA only when ≥80% of a continuous 5 min bout exceeded 100.6 mg.

## Adherence
We continuously monitored all participants' eating windows daily over the 12-week (84-day) intervention period. Adherence to flexTRE was computed as the percentage of days with an eating window of ≤ 8 h before 8 PM over the 12-week (84 days) intervention period. For the Exercise (EX) group, adherence was determined as the percentage of completed exercise sessions as required out of the total of 36 sessions (12 weeks * 3 sessions/week). For participants in the flexTRE +EX combination group, adherence was calculated separately for flexTRE and EX using the same criteria as above, and the overall adherence rate was determined by averaging the flexTRE and EX adherence values.

## Data analysis
Data were presented as mean ± standard deviation for continuous variables and as numbers or percentages for categorical variables. All randomized participants were included in the analysis according to Intention-to-treat principle. Missing data were not replaced, as the generalized estimating equations (GEE) model applies a natural and appropriate way to accommodate missing data[71]. Primary outcome, secondary outcomes, and the additional monitor parameters were analyzed by a GEE model using group and time as main effects and baseline values, age, menopausal status, and family history of cardiovascular disease included as covariates. Pairwise comparisons using linear contrast were performed to compare the differences between the intervention groups (flexTRE+EX vs. CON, flexTRE+EX vs. flexTRE, flexTRE+EX vs. EX, flexTRE vs. CON, and EX vs. CON) whenever a group × time interaction effect was detected. To account for multiplicity, the Holm-Bonferroni stepwise procedure was applied[72].

A subgroup analysis was conducted to explore whether the intervention effects differed based on menopausal status (pre-menopausal, peri-menopausal, and post-menopausal). Additionally, a per-protocol sensitivity analysis, restricted to participants with complete data across all assessments, was conducted to compare primary, secondary, and monitor parameters among groups. All statistical analyses were conducted using SPSS, version 28.0 (IBM Corp., Armonk, NY, USA), with significance set at $p < 0.05$.

**Reporting summary**

Further information on research design is available in the Nature Portfolio Reporting Summary linked to this article.

## Data availability

Source data for the individual-level changes in fat mass and body mass presented in Fig. 2 are included in the source data file. The study protocol is also provided in the supplementary Note 1. All other data are contained within the manuscript tables and supplementary information. To ensure participant confidentiality, additional de-identified participant data is not publicly deposited. However, the de-identified individual participant data that underlie the results reported in this article will be made available to researchers who provide a methodologically sound proposal for academic purposes. Data will be available beginning 6 months and ending 5 years following article publication. Proposals should be directed to hsswong@cuhk.edu.hk. To gain access, requestors will be required to sign a data access agreement. Source data are provided with this paper.

## Code availability

No previously unreported custom computer code was used in this manuscript.

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

## Acknowledgements

The authors gratefully acknowledge all the participants for their kind contributions to this study. The study was funded by an internal grant from the Chinese University of Hong Kong. The funding source had no role in study design, conduct, and analysis, or the decision to submit the manuscript for publication.

## Author contributions

Z.D. and S.H.W. oversaw the design, regulatory compliance, and execution of this study. Z.D., M.M., E.T.P., X.Y.T., C.H.S., and S.H.W. designed and conceptualized the study. Z.D., M.M., E.T.P., A.P.Y., and S.H.W. analyzed and interpreted the data. Z.D. and S.H.W. wrote and edited the manuscript. All the authors contributed to the composition and revision of the manuscript and gave final approval to its content.

## Competing interests

The authors declare no competing interests.

## Additional information

[1]Department of Sports Science and Physical Education, The Chinese University of Hong Kong, Shatin NT, Hong Kong SAR, China. [2]Faculty of Sport Sciences, Waseda University, Saitama, Japan. [3]School of Sport, Exercise and Health Sciences, Loughborough University, Leicestershire, UK. [4]School of Biomedical Sciences, Heart and Vascular Institute, Faculty of Medicine, The Chinese University of Hong Kong, Shatin NT, Hong Kong SAR, China. ✉e-mail: hsswong@cuhk.edu.hk

