## [Transparent Peer Review file · Nature Communications]

Flexible Time-Restricted Eating Combined with Exercise in a Free-Living Setting for Middle-Aged Women with Overweight/Obesity: A Randomized Controlled Trial

Corresponding Author: Professor Stephen Heung-Sang Wong

Version 0:

Reviewer comments:

Reviewer #1

(Remarks to the Author)

The study assess TRE and exercise, independently and combined, compared to a control on fat mass loss and cardiometabolic outcomes in middle aged women that have BMI in the overweight or obesity range. The study has clear findings that are interesting, but there are some methods and reporting issues that need to be addressed.

Minor comments

- The intro discusses post-menopausal women, but the average age is mid to late 40s which is pre- or peri-menopausal. The introduction should be adjusted accordingly
- Line 132: Do you mean differences in fat mass at baseline, or differences in the change of fat mass?
- Line 209. Specify the number of days that were used to calculate eating window and how it was calculated (ie. Window, daily average, etc)
- Line 210. I don't think discrepancy is the right word here. I would suggest difference.
- Line 215, don't start with no adverse events were reported, followed by except and a list of minor adverse events. Change it to say no serious adverse events were reported. Minor adverse events included...

Moderate issues/comments

- In the introduction, you need to clarify that the goal of the study was to compare all groups to control and the TRE and EX to the combined TRE+Ex.
 - o Is this what power calculations were done for?
 - o What was specified in the protocol and clinical trial registration?
- There needs to be a statistical test to assess differences between groups at baseline. Please add what was done to the legend in Table 1. If not differences, just mentioning it in the legend will suffice. If there were differences, it will need to be explained in the legend and noted in the table.
- Types of exercise that were chosen by participants should be reported.
- TRE and EX adherence need to be reported both separately and together for the combined group
- Need to describe how eating window was calculated.
- What tests were run to assess differences between groups at baseline? It seems like there were none. This needs to be done.

Major comments

- Exercise time at baseline and during/end of intervention needs to be reported for all groups. Baseline levels should be reported in table 1. The inclusion criteria was less than 150 mins per week of exercise, but the intervention is only 120 mins plus warm up and cool down (adding up to 150). It is important to see if exercise when from none to 150 or if it went from 100 to 120.
- It is unclear if participants had to maintain a consistent eating window, or if their eating window could change day to day as long as it was within 8am to 8pm. Please clarify. If consistent, adherence should be assessed based on the time of the personalized eating window (phase), not just the duration.
- This level of body composition analysis is usually done with a full DEXA scan. A TANITA bioelectrical impedance analyzer will give you the same outcomes, but not with the same reliability. A justification including references should be provided.

- o More details on the Tanita settings should be provided as the settings provided will change the report assessment
- The data analysis section does not have enough information. Specific tests used need to be provided.
- The study protocol looks like it is copied from the manuscript and does not include key information required for clinical trial protocols. Please explain.
- It seems that the control group and exercise groups did not have to monitor food timing. Why not?

Reviewer #2

(Remarks to the Author)

The authors present results of an interesting trial investigating the effects of time restricted eating (TRE) and physical exercise (EX) on body composition and metabolic health in middle-aged women. Findings suggest that combining time-restricted eating with aerobic exercise is an effective intervention to reduce fat mass and improve metabolic health. This could serve as a useful tool in the fight against the harmful body composition changes of middle-age. In combined TRE+EX group, significant changes in body composition were observed. The topic is timely and relevant, since it hasn't been previously studied in the context of middle-aged women, who are, as the authors state, at a high risk for metabolic disorders and several other chronic diseases.

Minor changes for the manuscript are suggested below:

Comment 1

Line 49: I would suggest leaving the word 'often' out from the sentence "As aging and menopause often coincide..".

Comment 2

In the manuscript, smokers were excluded from the study. However, this wasn't mentioned beforehand as an exclusion criteria in the Clinical Trial Registry. Are there data concerning the number of smokers, that were excluded from the study?

Comment 3

Line 313: When referring to women who have not yet gone through menopause, the use of word 'normal' feels slightly peculiar. Please use the word 'pre' instead, as it is common practise to use that word.

Comment 4

Line 313: The authors acknowledge in the discussion section that different menopausal statuses may have dissimilar impact on the results. Therefore it would be interesting to know, were the results different in different menopausal stages. Is the menopausal status of each participant known, and if yes, why not report that? In Study protocol paper menstrual cycle status was mentioned as a covariate in the Statistical Analysis section, yet it is unclear, whether this included the menopausal status (pre, peri or post).

Comment 5

Analysis strategy: Sample size estimation for this study is based on group-by-time interaction for the desired effect size, however, data analysis reports only group effects and post-hoc analysis for adjusted mean differences. Appropriate data interpretation needs to be based on significance of the group x time interaction terms. Please justify, why this was not the case. Furthermore, explain your choice of reporting/not reporting results based on intention to treat and per protocol procedures.

Comment 6

Adherence: Attrition rates are reported but not the adherence (although commented in the methods section) and how potential non-compliant participants were treated in the analysis.

Comment 7

Missing data: How did you handle missing data?

Reviewer #3

(Remarks to the Author)

Reviewer #4

(Remarks to the Author)

The authors present an RCT examining the combination of time-restricted eating and exercise vs either separate and vs neither. While the trial procedure and chosen outcomes are reasonable and well-documented, and the findings are interesting, there are problems with the analysis (especially the complete lack of control for multiple comparisons) and gaps in the presentation (in both the text of the manuscript and the analysis plan in the protocol) that need to be addressed.

In Figure 1, were outcomes followed for those who discontinued the intervention? If not, why not? If the goal was to test these regimens in a real-world setting, inability/unwillingness to maintain a regimen is part of that, and analyzing as

randomized (intent to treat), including everyone not lost to follow-up, makes the most sense, so if there is outcome data from those who discontinued the intervention, it would be good to see an as-randomized analysis. Also, whatever decisions were made around participants discontinuing their assigned regimen – whether to continue collecting outcome data and whether to include such participants in analyses – should have been decided prior to unblinding the data and described in the protocol.

The protocol notes that assessors were blinded but makes no mention of others. Did anyone involved in operational decisions have access to unblinded data while the trial was underway?

According to the analysis plan in the protocol, each outcome was examined across 5 pairwise between-group comparisons, but no adjustments for multiple comparisons were made, so the Type 1 error control is inadequate. In a clinical trial, Type 1 error is normally controlled at a family-wise error rate (FWER) of 5% across all comparisons involving the primary outcome(s). Using a simple Bonferroni correction, the significance threshold for comparison in the primary outcome should have been $.05/5 = .01$. The conclusions still appear to hold, but the treatment differences should be presented with 99% confidence intervals ($99\% = 1 - .01$).

The secondary outcomes are more problematic, as there are many of them and each one involves 5 comparisons. I suspect a purist might demand a 5% FWER across the entire set of secondary outcomes, but realistically, controlling at 5% separately among each category of outcomes seems reasonable. For other body composition parameters, there are 5 outcomes leading to 25 comparisons; a Bonferroni correction would put the significance threshold at $.05/25 = .002$, which would eliminate some differences in Table 2. (The authors could also consider a less conservative approach that still adequately controls the FWER such as the Hochberg step-up procedure.) Similarly, there are 25 comparisons among the 5 glycemic control outcomes, 50 comparisons among the 10 cardiometabolic markers, and 15 comparisons among the 3 questionnaires, so the necessary corrections will remove a number of significant findings across these outcomes. (The associated confidence intervals should also change; though looser than the proper significance thresholds, I'd be fine with 99% CIs since readers will find them more familiar and understandable than CIs like 99.8%, 99.9%, and 99.67%.)

Pre/post differences are presented with inferential tests in the tables, but those tests are not pre-specified in the protocol. At the very least, the significance tests would need to be adjusted for having 4 comparisons within each outcome; I leave the question of whether it is appropriate to include inferential analyses in this manuscript that were not pre-specified in the trial protocol to the editors.

“Energy intake and macronutrient composition” and “Eating window change” are in the results under secondary outcomes, but only the former is described in the methods, and neither appears in Table 3 or in the protocol. The main findings paper for a clinical trial normally only includes pre-specified outcomes and analyses notes in the protocol or a statistical analysis plan.

It would be helpful to provide the MCID (minimal clinically important difference) for all the measures where one is known to help the reader decide if the statistically significant differences are also clinically meaningful.

The captions for Figures 2 and 3 should provide more information. Are the error bars confidence intervals (and if so, what width), standard deviations, standard errors, or something else? Also, all abbreviations used should be explained in the legend for each figure they appear in even if they're also defined in the text. For Figure 3, plotting the group differences and their confidence intervals would probably be more useful than plotting the group means (especially since the pre, post, and change scores are presented in tables for the other outcomes). But as a more general question, why are these two secondary/exploratory (it's unclear as noted above) outcomes given figures while the primary and main secondary outcomes are not? That seems odd, since a reader would assume the figures show what the authors consider the most important findings of the paper.

I believe these issues can all be addressed, and it looks like your primary findings (though not all of the secondaries) will survive the necessary adjustments for multiple comparisons. I look forward to seeing the next iteration.

Version 1:

Reviewer comments:

Reviewer #1

(Remarks to the Author)

Thank you for your revision. I believe they have strengthened the paper. There are still a couple of issue that need to be resolved.

Major:

1. Thank you for clarifying how the TRE intervention was done. However, the flexible TRE window needs to be mentioned throughout the paper as allowing a flexible window contradicts the primary component of TRE. TRE requires a consistent eating window as defined by the International Consensus of Fasting ([https://www.cell.com/cell-metabolism/fulltext/S1550-4131\(24\)00269-9](https://www.cell.com/cell-metabolism/fulltext/S1550-4131(24)00269-9)). To be more accurate, it should be referred to as 16:8 intermittent fasting which allows the eating window to move. However, if the authors insist on calling it TRE, then in the title, abstract, intro, and discussion, it needs to be made

clear that this is modified form of TRE. Examples including calling it flexible TRE (flexTRE) or modified TRE (mTRE), but it should not be referred to as TRE alone as it is very misleading.

Minor:

1. In the abstract and introduction, you should include the duration of the TRE interval (ex: 8 hour TRE) and when they were allowed to eat (set time or personalized)
2. Figure 1 quality needs to be improved. This may just be because it is in the manuscript.
3. Figure 2: the axis should have the same scale for a given outcome. Example: the axis scale for all fat mass changes should be the same and all body mass change should be the same so the reader can easily see the differences
4. It's interesting that fat-free mass only decreased in the TRE+EX group and not in TRE or EX alone. How do you explain this? It would be good to add this to the discussion.

Reviewer #2

(Remarks to the Author)

I thank the authors for thoroughly responding to my review report. I do not have further concerns or comments.

Reviewer #3

(Remarks to the Author)

Reviewer #5

(Remarks to the Author)

Thank you for the opportunity to review the revised version of the manuscript.

I have carefully examined the authors' responses and the updated manuscript. I am pleased to note that all previous comments of reviewer 4 have been adequately addressed. The revised version has improved the clarity and quality of the work, and I only have two minor comments:

1. The abbreviation "CON" in the abstract appears without an initial definition. For clarity, please provide the full term at its first mention, followed by the abbreviation in parentheses, e.g. control (CON).
2. To assist readers who may wish to explore the technical details, please consider citing the following reference as the original source introducing the Holm-Bonferroni method.
Holm, S. (1979). "A simple sequentially rejective multiple test procedure". *Scandinavian Journal of Statistics*. 6 (2): 65–70.

REVIEWER COMMENTS

Reviewer #1 (Remarks to the Author):

The study assess TRE and exercise, independently and combined, compared to a control on fat mass loss and cardiometabolic outcomes in middle aged women that have BMI in the overweight or obesity range. The study has clear findings that are interesting, but there are some methods and reporting issues that need to be addressed.

We thank the reviewer for the thoughtful feedback and for highlighting the interesting findings of our study. We appreciate the opportunity to address the methodological and reporting issues raised, and we have carefully considered all comments and suggestions to improve the clarity and rigor of our manuscript. Below, we provide detailed responses to each point raised.

Minor comments

- The intro discusses post-menopausal women, but the average age is mid to late 40s which is pre- or peri-menopausal. The introduction should be adjusted accordingly

Reply: Thank you for highlighting this important point. We appreciate your observation that the average age of our study population is in the mid to late 40s, placing participants primarily in a pre- or peri-menopausal stage rather than strictly post-menopausal. To address your comment, we have revised the second paragraph in the Background to emphasize the significance of the menopausal transition in midlife women:

“Given that cardiovascular disease remains the leading cause of death among women, the menopausal transition period typically beginning in women's 40s and progressing from pre- through peri- to post-menopause, represents a critical window for intervention, as it is characterized by a significant increase in cardiovascular risk. Effective weight management strategies and early preventive interventions are crucial due to the unfavorable alterations in body composition and metabolic profile in midlife women.” [Page 2, Ln 53-58]

These changes ensure that our introduction accurately represents the menopausal status of our participants and highlights the importance of early preventive interventions during this critical life stage. We believe this revision more clearly sets the context for our study and better supports its relevance to midlife women's health. Thank you once again for your feedback.

- Line 132: Do you mean differences in fat mass at baseline, or differences in the change of fat mass?

Reply: Thank you for pointing out the ambiguity in our description regarding fat mass differences noted in line 132. It means the differences in the change in fat mass, other than mean differences in fat mass at baseline. We apologize for the confusion.

To clarify: “A group-by-time interaction effect was observed in fat mass ($P < 0.001$). Compared with the control group, all intervention groups including TRE, EX, and TRE+EX showed a larger reduction in fat mass ...” [Page 5, Ln 143-144]

We have revised the relevant section to make this statement clearer and ensure readers understand the exact nature of the comparisons being made. We appreciate your feedback and believe this update provides the necessary clarity.

- Line 209. Specify the number of days that were used to calculate eating window and how it was calculated (ie. Window, daily average, etc)

Reply: Thank you for your comment regarding the calculation of the eating window.

We have revised the section to specify the precise methodology: “Participants were instructed to record their daily eating window for one week prior to the intervention (baseline) and throughout the 12-week (84-day) intervention. The daily eating window was defined as the interval between the first and last reported food intake of each day. To assess behavior change, eating window data from the seven days before the intervention and the final seven days of the intervention were compared.” [Page 15, Ln 555-559]

- Line 210. I don't think discrepancy is the right word here. I would suggest difference.

Reply: Thank you for pointing out our wording choice in line 210. We agree that using “difference” will be more accurate in expressing our meaning. We have revised the sentence here “A group-by-time interaction effect was observed in eating window ($P < 0.001$). The TRE and TRE+EX groups exhibited significantly greater reductions in eating windows compared to the EX and CON group.” to avoid confusion and maintain clarity in the manuscript. [Page 7, Ln 225-227]

- Line 215, don't start with no adverse events were reported, followed by except and a list of minor adverse events. Change it to say no serious adverse events were reported. Minor adverse events included...

Reply: Thank you for noting the wording issue related to adverse events. We agree with your suggestion and have revised the paragraph accordingly: “No serious adverse events were reported throughout the study duration. Minor adverse events included heightened feelings of hunger that might have impacted sleep in six participants undergoing the TRE and TRE+EX interventions during the initial two weeks. Additionally, two individuals in the EX group reported slight discomfort and tiredness during the first two weeks of exercise sessions, expressing difficulty in completing the entire session. These minor adverse events were primarily observed during the early stages of the intervention and subsided as participants adapted to the new regimens.” [Page 8-9, Ln 254-260]

We appreciate your feedback and believe this new phrasing better aligns with your recommendation.

Moderate issues/comments

- In the introduction, you need to clarify that the goal of the study was to compare all groups to control and the TRE and EX to the combined TRE+Ex.

Reply: In our original manuscript, we stated the purpose as follows: “Therefore, this study aims to investigate the isolated and combined effects of TRE and aerobic exercise on body

composition and metabolic health outcomes in this specific population in a free-living setting.” However, we recognize that this wording did not explicitly mention comparisons of each intervention arm with the control group, nor comparisons between single and combined interventions. Based on your recommendation, we have revised the purpose statement to read:

Revised Purpose Statement: “Therefore, this study aims to investigate whether TRE and aerobic exercise (EX), alone or in combination (TRE+EX), improve body composition and metabolic health outcomes relative to passive control in this specific population in a free-living setting. Specifically, we seek to evaluate whether the combined intervention (TRE+EX) produces superior outcomes compared to individual interventions (TRE or EX). Through these comparisons, we aim to elucidate the individual and synergistic impacts of these interventions on health outcomes, providing valuable insights for effective lifestyle modification strategies.” [Page 3, Ln115-122]

o Is this what power calculations were done for?

Reply: We appreciate your inquiry about our power calculations and acknowledge that our initial description of the study purpose may have created confusion.

Our original sample size calculation was based on an estimated effect size ($f \approx 0.25$) derived from previous research comparing alternate-day fasting (ADF), exercise, ADF+EX, and control in a four-arm randomized controlled trial (RCT). This prior study also investigated the combined effects of intermittent fasting and endurance exercise. Our initial plan focused on four key comparisons: (1) TRE vs. CON, (2) TRE+EX vs. CON, (3) TRE vs. TRE+EX, and (4) EX vs. TRE+EX. However, as we refined our analysis plan, we decided to include an additional comparison (5) EX vs. CON—to provide a more comprehensive understanding of the intervention effects.

During the re-evaluation of our power analysis, we recognized that the EX vs. CON comparison might exhibit a smaller effect size ($f \approx 0.18$) than initially anticipated. Since this could reduce the power to detect significant differences, we adjusted our calculations to account for this smaller effect size. To ensure feasibility while maintaining adequate power, we set our power level at 80% ($\alpha=0.05$) for all five comparisons.

To account for multiple comparisons, we applied a stricter significance threshold ($\alpha=0.01$) to test if our sample size fulfills the requirement for each compare groups. For example, detecting a significant interaction effect ($f=0.63$) between COM and CON with 80% power and $\alpha=0.01$ would require a minimum of 12 participants per group. Similarly, for comparisons such as ADF vs. CON ($f=0.25$), COM vs. ADF ($f=0.38$), and COM vs. EX ($f=0.71$), a minimum of 52 participants across the two comparison groups would be required. Based on these calculations, our total sample size of 104 participants (26 per group) is sufficient to ensure adequate power for all planned comparisons, even considering multiple testing and an anticipated 10% dropout rate.

In the revised Methods section, we have included a detailed explanation of the recalculated sample size, specifying how including all five comparisons—particularly EX vs. CON—led to

an increased estimate. "Sample size estimation was conducted using G*Power software based on an effect size derived from a previous study examining the combined effects of ADF with endurance exercise in adults with obesity. Based on these data, an effect size for the group-by-time interaction was calculated (Effect size $f = 0.18$). Using this effect size, a power analysis determined that 23 participants per group would be required to achieve 80% statistical power at a two-sided significance level of $\alpha = 0.05$. To account for an anticipated dropout rate of 10%, the study aimed to enroll a total of at least 102 participants, with 26 participants allocated to each of the four groups. The power analysis was reperformed to generate a more conservative sample size estimation, ensuring an adequate sample size for the predesignated pairwise comparisons." [Page 13, Ln 438-446]

By ensuring sufficient power for all comparisons, including EX vs. CON, we have strengthened our study design to robustly investigate and compare the isolated and combined effects of TRE and EX relative to a passive control on body composition and metabolic health outcomes.

(Reference: Overall four group interaction $f=0.56$, COM vs. CON= 0.63 , ADF vs. CON= 0.25 , EX vs. CON= 0.18 , COM vs. ADF= 0.38 , COM vs. EX= 0.71).

o What was specified in the protocol and clinical trial registration?

Reply: In the original protocol and clinical trial registration, we stated that "this study aims to investigate the isolated and combined effects of TRE and aerobic exercise on body composition and metabolic health outcomes in this specific population in a free-living setting." However, our statement may create confusion. We have therefore revised our stated purpose to clarify the all five pairwise comparisons and recalculated the required sample size accordingly. This ensures that our final analyses align with both our initial aims and the additional comparisons identified during the study.

- There needs to be a statistical test to assess differences between groups at baseline. Please add what was done to the legend in Table 1. If not differences, just mentioning it in the legend will suffice. If there were differences, it will need to be explained in the legend and noted in the table.

Reply: Thank you for your helpful feedback regarding group comparisons at baseline. We have conducted the statistical tests to assess differences between groups at baseline and revised this section as: "Basic characteristics about age, height, body mass, body mass index (BMI), body fat percentage, fat mass, fat free mass, waist circumference, moderate-to-vigorous physical activity per week (MVPA), eating window, daily energy intake, and macronutrient distribution (carbohydrate, protein, and fat percentage), menopausal status, and family history of cardiovascular disease are detailed in Table 1, no significant differences among groups were observed." [Page 3-4, Ln 129-134]

However, according to the CONSORT statement, we didn't include the results of baseline significance testing in the manuscript. We appreciate your suggestion and believe this revision enhances the clarity of our methods.

- Types of exercise that were chosen by participants should be reported.

Reply: Thank you for your valuable suggestion to include details on the types of exercise chosen by participants. As part of the EX and TRE+EX group exercise protocol, participants were instructed to engage in aerobic exercise three times per week, aiming to achieve moderate intensity based on their maximum heart rate. Participants were given the flexibility to choose from running, jogging, or brisk walking, either indoors or outdoors. To classify the reported exercise types, we referred to participants' exercise diaries, which included their average speeds during exercise. Following the CDC measuring physical activity intensity and other health organizations, we defined brisk walking as 3–4 miles per hour, jogging as 4–6 miles per hour, and running as 6–8 miles per hour.

We have provided the piece of information in the adherence rate part: “In addition, participants in the EX group and TRE+EX group engaged in aerobic exercise as prescribed, with flexibility to choose from brisk walking, jogging, or running, ensuring the prescribed heart rate was achieved. In the EX group (n = 24), 6 participants (25.0%) chose brisk walking, 11 participants (45.8%) chose jogging, and 4 participants (16.7%) chose running. 2 participants (8.3%) opted to use an elliptical machine rather than running-related activities, as they reported discomfort or difficulty within the first two weeks. Additionally, one participant (4.2%) didn't do any exercise as prescribed during the intervention period. In the TRE+EX group (n = 25), 1 participant (4.0%) performed brisk walking, 19 participants (76.0%) performed jogging, and 5 participants (20.0%) performed running.” [Page 9, Ln 270-278]

- TRE and EX adherence need to be reported both separately and together for the combined group

Reply: Thank you for highlighting the importance of reporting adherence to the intervention protocols both separately and jointly for the combined group. In response to your comment, we have updated the manuscript to include detailed adherence data for each group over the 12-week period.

We have provided this information: “Participants in the TRE+EX group achieved an average adherence rate of 89.3% for TRE and 79.6% for EX, resulting in an overall average adherence rate of 84.6% for this group.” [Page 9, Ln 267-268 & Figure 3, graph D]

- Need to describe how eating window was calculated.

Reply: Thank you for pointing out the need to clarify how the eating window was calculated. We continuously monitored participants' eating windows daily over the 12-week (84-day) intervention period. For each participant, the daily eating window was calculated as the time between the first and last reported food intake of the day. We have included this part of the information in the method part in monitor parameters and we revised the manuscript as: “Participants were instructed to record their daily eating window for one week prior to the intervention (baseline) and throughout the 12-week (84-day) intervention. The daily eating window was defined as the interval between the first and last reported food intake of each day. To assess behavior change, eating window data from the seven days before the intervention and the final seven days of the intervention were compared. The records from the entire 12-week period also served to evaluate adherence to the prescribed eating window.” [Page 15, Ln555-560]

- What tests were run to assess differences between groups at baseline? It seems like there were none. This needs to be done.

Reply: Thank you for pointing out the need to assess differences between groups at baseline. We agree that this is an important step to ensure comparability across groups. To address this concern, we conducted statistical analyses to compare baseline characteristics between groups. Specifically, we used one-way ANOVA for continuous variables, and Chi-square tests for categorical variables assess any significant differences. Based on these analyses, we found no significant differences between groups at baseline for key demographic or clinical variables. Please find the attached table below, which provide our results regarding the baseline group differences test. And we didn't observe any significant difference among groups.

	TRE (n=26)	EX (n=26)	TRE+EX (n=26)	CON (n=26)	P value
Age, years	46 ± 5	49 ± 6	48 ± 6	47 ± 6	0.117
Height, cm	159.3 ± 5.8	158.5 ± 6.1	160.6 ± 5.4	158.9 ± 4.8	0.546
Body mass, kg	69.7 ± 7.6	67.2 ± 7.9	68.3 ± 8.9	69.2 ± 11.0	0.769
Body mass index, kg/m ²	27.6 ± 3.6	26.9 ± 2.6	26.3 ± 2.7	27.2 ± 3.5	0.470
Body fat percentage, %	38.5 ± 5.5	38.3 ± 3.7	37.0 ± 3.8	38.8 ± 5.4	0.504
Fat mass, kg	27.2 ± 6.7	26.0 ± 5.2	25.5 ± 6.1	27.4 ± 9.1	0.704
Fat free mass, kg	42.6 ± 2.8	41.3 ± 3.4	42.7 ± 3.3	41.9 ± 3.2	0.326
Waist circumference, cm	89.5 ± 6.7	87.9 ± 5.5	87.6 ± 8.3	90.4 ± 9.3	0.491
MVPA, min/week	61.4 ± 58.9	78.0 ± 62.9	61.7 ± 60.8	57.5 ± 72.6	0.664
Eating window, h/day	10.4 ± 1.1	10.5 ± 1.4	10.6 ± 1.7	11.0 ± 1.5	0.558
Energy intake, kJ/day	7173.6 ± 587.6	7133.1 ± 452.6	7100.3 ± 387.2	6907.4 ± 749.1	0.333
Carbohydrate percentage, %	0.45 ± 0.06	0.47 ± 0.08	0.44 ± 0.06	0.42 ± 0.07	0.117
Protein percentage, %	0.18 ± 0.04	0.17 ± 0.03	0.17 ± 0.03	0.19 ± 0.04	0.171
Fat percentage, %	0.37 ± 0.05	0.36 ± 0.07	0.39 ± 0.06	0.39 ± 0.05	0.163
Menopausal status, n (%)					0.131
Premenopausal	16 (61.5%)	10 (38.5%)	17 (65.4%)	15 (57.7%)	
Perimenopausal	6 (23.1%)	3 (11.5%)	2 (7.7%)	4 (15.4%)	
Postmenopausal	4 (15.4%)	13 (50.0%)	7 (26.9%)	7 (26.9%)	
Family history of cardiovascular disease, n (%)					0.797
With family history	17 (65.4%)	19 (73.1%)	16 (61.5%)	16 (61.5%)	
No family history	9 (34.6%)	7 (26.9%)	10 (38.5%)	10 (38.5%)	

However, according to the CONSORT statement, one of the recommendations is that a table is presented showing baseline demographic and clinical characteristics for each group. And according to their recommendation: “significance testing of baseline differences in randomized controlled trials should not be performed.” Therefore, while we acknowledge the importance of these analyses to confirm comparability, we have elected not to include the results of baseline significance testing in the manuscript. Instead, we have clarified in the Results section that “no significant differences among groups were observed” [Page 4, Ln133-134].

<https://www.bmj.com/content/340/bmj.c869>

<https://doi.org/10.1186/s12966-015-0162-z>

We hope this explanation adequately addresses your concern while aligning with best practices for reporting randomized controlled trials.

Major comments

- Exercise time at baseline and during/end of intervention needs to be reported for all groups. Baseline levels should be reported in table 1. The inclusion criteria was less than 150 mins per week of exercise, but the intervention is only 120 mins plus warm up and cool down (adding up to 150). It is important to see if exercise when from none to 150 or if it went from 100 to 120.

Reply: Thank you for raising this important concern regarding baseline and intervention exercise levels. We fully agree that reporting exercise time at baseline and during/end of the intervention is crucial for understanding changes in physical activity levels across groups. As you pointed out, it is particularly important to clarify whether exercise time increased from none to 150 minutes per week or from a higher baseline level (e.g., 100 to 120 minutes per week).

To address this concern, we provided the physical activity data collected using ActiGraph accelerometers, which provide objective and reliable measurements of moderate-to-vigorous physical activity (MVPA). This approach ensures greater accuracy compared to self-reported questionnaires, minimizing bias, and providing robust data on participants' physical activity levels. We have incorporated this information into the manuscript as follows:

Baseline Data: Participants in all four groups had an average baseline MVPA of approximately 65 minutes per week, which aligns with the study's inclusion criteria (<150 minutes per week). This information has been added to Table 1 under baseline characteristics, providing a clear depiction of participants' physical activity levels before the intervention.

End-of-Intervention Data: At the end of the intervention, the TRE and CON groups maintained similar weekly MVPA levels (63 and 62 minutes per week, respectively). In contrast, the EX and TRE+EX groups demonstrated substantial increases in MVPA, reaching 209 and 211 minutes per week, respectively. This indicates that the EX and TRE+EX interventions successfully elevated participants' physical activity levels beyond the recommended threshold of 150 minutes per week, while the TRE and CON groups experienced no significant changes. These results have been detailed in Table 3 under additional monitor parameters.

Methods Section: The manuscript now describes the use of ActiGraph accelerometers to assess MVPA in the Methods section, under the "Monitor Parameters" subsection as follow: "During the week prior to intervention and the final week of the intervention, participants were asked to wear an ActiGraph GT3X-BT accelerometer (ActiGraph Corp, Pensacola, FL, USA) on their nondominant wrist for seven consecutive days (baseline and post-intervention). Participants removed the device only during water-based activities (e.g.,

showing, bathing, swimming) and recorded the non-wearing times and reasons in an activity log. The accelerometer recorded raw acceleration along three axes at a sampling frequency of 100 Hz, producing files in .gt3x format. These files were downloaded using ActiLife software (ActiGraph Corp, Pensacola, FL, USA) and subsequently analyzed in R (4.4.1) with the GGIR package. Time spent in MVPA was defined by a mean acceleration of ≥ 100.6 mg, based on a laboratory calibration study of adults aged 21 – 61 years. To reduce the likelihood of misclassifying brief artifacts as MVPA, activity was further classified as MVPA only when $\geq 80\%$ of a continuous 5-minute bout exceeded 100.6 mg.” This provides clarity on how MVPA was objectively measured throughout the study. [Page 15-16, Ln 561-571]

We believe these additions significantly enhance the manuscript by providing a more comprehensive understanding of participants’ physical activity patterns at baseline and during the intervention. This addresses the reviewer’s concern and strengthens the interpretation of our findings, particularly the impact of the intervention on exercise behavior. Thank you again for highlighting this critical point. Please let us know if further clarifications or analyses are required.

- It is unclear if participants had to maintain a consistent eating window, or if their eating window could change day to day as long as it was within 8am to 8pm. Please clarify. If consistent, adherence should be assessed based on the time of the personalized eating window (phase), not just the duration.

Reply: Thank you for highlighting this important point. Participants were not required to adhere to a fixed eating window throughout the intervention. Instead, they were given the flexibility to select their preferred eating window each day, provided it remained within the range of 8:00 AM to 8:00 PM. This flexible approach was intentionally chosen to reflect real-world conditions, thereby increasing the practicality and feasibility of adhering to the protocol.

We have revised the manuscript to clarify this: “Participants had the flexibility to select their preferred eating window, starting anytime between 8:00 AM and 12:00 PM and ending between 4:00 PM and 8:00 PM. To better reflect real-world conditions, participants were allowed to adjust their eating window on a day-to-day basis as long as it remained within the 8:00 AM to 8:00 PM range. Consistency in the specific timing of the eating window was not required.” [Page 13, Ln 459-463]

While we acknowledge that adherence to a consistent eating window could provide additional insights, we believe this flexible approach enhances the ecological validity of the intervention and better represents its application in real-world scenarios. We hope this addresses your concern and provides greater clarity. Please let us know if additional details are needed.

- This level of body composition analysis is usually done with a full DEXA scan. A TANITA bioelectrical impedance analyzer will give you the same outcomes, but not with the same reliability. A justification including references should be provided.

Reply: Thank you for raising this important concern regarding the use of the TANITA bioelectrical impedance analyzer (BIA) for body composition analysis. We fully acknowledge that dual-energy X-ray absorptiometry (DEXA) is widely regarded as the gold standard for body composition measurements. However, we have reviewed the literature and found evidence supporting the validity of the TANITA BIA when compared to DEXA. Specifically, a study by María et al. (2021) demonstrated that the TANITA BIA achieves excellent agreement with DEXA for fat mass measurements, reporting an intraclass correlation coefficient (ICC) of 0.925 for fat mass (María et al., 2021, <https://doi.org/10.1016/j.nut.2021.111442>). These findings confirm the reliability of the TANITA BIA for assessing body composition in research contexts.

To incorporate this justification into the manuscript, we have revised the relevant section as follows: “Fat mass was evaluated following an 8-hour fast avoiding the consumption of liquids with a bioelectrical impedance analyzer (MC-780 MA, Tanita, Japan) that has been previously validated compared to dual-energy x-ray absorptiometry [fat mass (kg) ICC: 0.925].” [Page 14, Ln 489-491]

We believe this addition strengthens the manuscript by providing a clear justification for the use of the TANITA BIA and referencing its validation against the gold standard DEXA. We appreciate your feedback and welcome any further suggestions.

o More details on the Tanita settings should be provided as the settings provided will change the report assessment

Reply: Thank you for your comment. We have ensured that all measurements using the Tanita bioelectrical impedance analyzer (BIA) were conducted following a standardized protocol to ensure consistency and accuracy.

Specifically, the protocol was as follows: “Participants wore light, form-fitting attire and remained barefoot, with 0.5 kg consistently deducted to account for their clothing. Prior to each measurement, we entered the participants’ date of birth, gender, and height into the analyzer, selected the “normal” body type (rather than “athlete”), and applied the Asian judgment setting. Body mass index (BMI) was calculated according to the World Health Organization (WHO) guidelines, ensuring a uniform and standardized assessment across all tests.” In addition, we asked the participants to conduct overnight fasting for at least 8 hours and avoid the consumption of liquids before the measurement. [Page 14, Ln 493-498]

These details have been added to the manuscript to provide greater clarity on the methodology and ensure transparency regarding the settings used during the assessment. We hope this addresses your concern. Please let us know if additional information is needed.

- The data analysis section does not have enough information. Specific tests used need to be provided.

Reply: Thank you for your comment. We have carefully reviewed the data analysis section and agree that additional details about the specific statistical tests used would enhance the clarity of our manuscript.

To address this, we have revised the data analysis section to include the following information: “Data were presented as mean \pm standard deviation for continuous variables and as numbers or percentages for categorical variables. All randomized participants were included in the analysis according to Intention-to-treat principle. Missing data were not replaced, as the generalized estimating equations (GEE) model applies a natural and appropriate way to accommodate missing data. Primary outcome, secondary outcomes, and the additional monitor parameters were analyzed by a GEE model using group and time as main effects and baseline values, age, menopausal status, and family history of cardiovascular disease included as covariates. Pairwise comparisons using linear contrast were performed to compare the differences between the intervention groups (TRE+EX vs. CON, TRE+EX vs. TRE, TRE+EX vs. EX, TRE vs. CON, and EX vs. CON) whenever a group \times time interaction effect was detected. To account for multiplicity, the Holm-Bonferroni stepwise procedure was applied.

A subgroup analysis was conducted to explore whether the intervention effects differed based on menopausal status (pre-menopausal, peri-menopausal, and post-menopausal). Additionally, a per-protocol sensitivity analysis, restricted to participants with complete data across all assessments, was conducted to compare primary, secondary, and monitor parameters among groups. All statistical analyses were conducted using SPSS, version 28.0 (IBM Corp., Armonk, NY, USA), with significance set at $p < 0.05$.” [Page 16, Ln 582-598]

- The study protocol looks like it is copied from the manuscript and does not include key information required for clinical trial protocols. Please explain.

Reply: We sincerely appreciate the reviewer's attention to this important detail regarding our study protocol. Below we clarify the key points:

The similarity between the protocol and manuscript exists because the journal requires submission of the original approved protocol, and our methods section strictly reflects this pre-approved study design. Consequently, some sections of both documents appear similar, which may have created the impression of duplication, though this was unintentional.

We acknowledge that certain elements typically included in clinical trial protocols, such as detailed data management procedures and participant risk mitigation, were not included in the submitted document. This is because our institution's ethics review process involves submitting such information through separate routes in the IRB application system. These components were thoroughly reviewed and approved by the ethics committee but were not consolidated into the main protocol document. When conducting the study, we strictly followed standard clinical trial procedures regarding recruitment, eligibility screening, informed consent, randomization, blinding, and data management according to established guidelines and ethical requirements.

We appreciate the reviewer's valuable feedback, which has helped us clarify and improve the transparency of our submission.

- It seems that the control group and exercise groups did not have to monitor food timing. Why not?

Reply: Thank you for raising this important question. Participants in the exercise and control groups were instructed to maintain their habitual diet and eating patterns without specific monitoring of food timing. This decision was made to preserve their natural dietary behavior, avoid introducing monitoring that could alter habits, and provide a realistic comparison to the time-restricted eating (TRE) intervention. By not imposing specific eating windows, we ensured the ecological validity of the study and strengthened the generalizability of our findings. Self-reported eating windows were collected from all groups to document eating behaviors for context. We hope this clarifies our rationale and study design.

Reviewer #2 (Remarks to the Author):

The authors present results of an interesting trial investigating the effects of time restricted eating (TRE) and physical exercise (EX) on body composition and metabolic health in middle-aged women. Findings suggest that combining time-restricted eating with aerobic exercise is an effective intervention to reduce fat mass and improve metabolic health. This could serve as a useful tool in the fight against the harmful body composition changes of middle-age. In combined TRE+EX group, significant changes in body composition were observed. The topic is timely and relevant, since it hasn't been previously studied in the context of middle-aged women, who are, as the authors state, at a high risk for metabolic disorders and several other chronic diseases.

We sincerely thank the reviewer for their positive and encouraging feedback on our study. We are pleased that the relevance, and potential impact of our findings were recognized, particularly in the context of addressing the metabolic health challenges faced by middle-aged women. We greatly appreciate your thoughtful comments and have carefully addressed all points raised in our responses below.

Minor changes for the manuscript are suggested below:

Comment 1

Line 49: I would suggest leaving the word 'often' out from the sentence "As aging and menopause often coincide..".

Reply: We appreciate the reviewer's suggestion. We agree that removing the word "often" makes the sentence more concise and precise. Accordingly, we have revised the sentence: "As aging and menopause coincide...". Thank you for pointing this out. [Page 2, Ln 50]

Comment 2

In the manuscript, smokers were excluded from the study. However, this wasn't mentioned beforehand as an exclusion criteria in the Clinical Trial Registry. Are there data concerning the number of smokers, that were excluded from the study?

Reply: We sincerely apologize for the oversight in not specifying "smokers" as an exclusion criterion in the Clinical Trial Registry. Smoking has always been one of the exclusion criteria for our study, as outlined in both the manuscript and the study protocol to ensure consistency. Unfortunately, this criterion was inadvertently omitted from the Clinical Trial

Registry due to an error during the registration process. To address this issue, we have updated the Clinical Trial Registry to include this exclusion criterion, ensuring alignment with our protocol and greater clarity moving forward.

We also confirm that no smokers were excluded from the study, as none of the voluntary participants reported smoking during the screening process. We appreciate the reviewer for bringing this to our attention and allowing us to clarify this matter.

Comment 3

Line 313: When referring to women who have not yet gone through menopause, the use of word 'normal' feels slightly peculiar. Please use the word 'pre' instead, as it is common practice to use that word.

Reply: Thank you for highlighting this point. We agree that the term "pre" is more appropriate and aligns with common practice. We have revised the text to replace the word "normal" with "pre" when referring to women who have not yet gone through menopause. Accordingly, we have revised the sentence: "First, the inclusion of participants at various menopausal stages (pre, peri-, post-) may introduce physiological variability." [Page 11, Ln 382-383]

We appreciate your suggestion, which enhances the accuracy and clarity of the manuscript.

Comment 4

Line 313: The authors acknowledge in the discussion section that different menopausal statuses may have dissimilar impact on the results. Therefore it would be interesting to know, were the results different in different menopausal stages. Is the menopausal status of each participant known, and if yes, why not report that? In Study protocol paper menstrual cycle status was mentioned as a covariate in the Statistical Analysis section, yet it is unclear, whether this included the menopausal status (pre, peri or post).

Reply: Thank you for raising this important point. We have included data on menopausal status (pre-, peri-, and post-menopausal) in the baseline characteristics, which are presented in Table 1. As mentioned in our study protocol, we collected each participant's menopausal status and incorporated it as a covariate in the statistical analysis.

To further examine the potential impact of menopausal status on the results, we conducted an additional analysis to investigate whether there was a significant interaction effect of time × group × menopausal status on the primary and secondary outcomes. Significant interaction effects were observed for several outcomes. As a result, we performed subgroup analyses to evaluate the intervention effects within each menopausal status group (pre-, peri-, and post-menopausal). These detailed findings have been included in the result part as: "We conducted a subgroup analysis based on the participants' menopausal status. When a significant interaction effect between menopausal status-by-group-by-time was identified, we further analyzed this outcome within subgroups. Detailed results of this analysis are provided in Supplementary Table S1." [Page 8, Ln 244-247]. These findings were also included in the discussion part and supplementary Table S1.

However, we acknowledge the unequal distribution of participants across menopausal stages, with 58 participants in the premenopausal group, 15 in the perimenopausal group, and 31 in the postmenopausal group. This imbalance, especially after participants were divided into the four intervention groups, resulted in smaller sample sizes within each subgroup. For example, some results in the perimenopausal group showed adverse effects compared to the other two groups, but it is difficult to determine whether these findings reflect true differences or are due to the small sample size, which limits the ability to detect meaningful changes. However, it serves as a pilot analysis. We have further mentioned this in the limitation as “Although subgroup analyses were conducted, the imbalance and small sample size among groups limit the robustness of these conclusions.” [Page 11, Ln 383-385]

We agree that further studies with larger and more balanced sample sizes are needed to confirm whether menopausal status influences the effects of the intervention. We appreciate your valuable comment, which has allowed us to improve the analysis and provide a more comprehensive discussion of our findings.

Comment 5

Analysis strategy: Sample size estimation for this study is based on group-by-time interaction for the desired effect size, however, data analysis reports only group effects and post-hoc analysis for adjusted mean differences. Appropriate data interpretation needs to be based on significance of the group x time interaction terms. Please justify, why this was not the case. Furthermore, explain your choice of reporting/not reporting results based on intention to treat and per protocol procedures.

Reply: We appreciate your valuable feedback. We acknowledge that our original sample size calculation was based on the group-by-time interaction for the desired effect size, whereas our initial data analysis focused primarily on group effects and post-hoc comparisons of adjusted mean differences. In response to your suggestion, we have revised our analyses to explicitly include the group-by-time interaction terms, ensuring that the results directly reflect the methodological rationale underlying our sample size determination.

Notably, our use of generalized estimating equations (GEE) already accounts for baseline values as covariates to capture changes between time points. The outcomes of the group-by-time interaction analyses are consistent with our previous group-focused findings, reinforcing the robustness of our initial conclusions. However, we believe it is crucial to present the results in a manner that is more transparent and accessible to readers, which is why we now place greater emphasis on these interaction effects.

In addition, we have conducted both intention-to-treat (ITT) and per-protocol (PP) analyses to provide a comprehensive assessment of our intervention’s impact. Presenting both ITT and PP findings enhances the rigor and transparency of our study by demonstrating how the inclusion or exclusion of participants with protocol deviations may influence the observed effects. We have updated the ITT results in Table 2 and Table 3. The PP results have been summarized in Supplementary Table S2.

We appreciate your comment and the opportunity to clarify and strengthen our analysis strategy and its presentation in the manuscript.

Comment 6

Adherence: Attrition rates are reported but not the adherence (although commented in the methods section) and how potential non-compliant participants were treated in the analysis.

Reply: Thank you for highlighting the need for more detailed reporting on adherence and the treatment of non-compliant participants. While the adherence rate was briefly mentioned in our Discussion section, we did not provide sufficient details on how it was calculated.

Specifically, we have expanded the Results and Method section to explain precisely how adherence was determined. In method: “We continuously monitored participants’ eating windows daily over the 12-week (84-day) intervention period. Adherence to TRE was computed as the percentage of days with an eating window of ≤ 8 hours before 8 PM over the 12-week (84 days) intervention period. For the Exercise (EX) group, adherence was determined as the percentage of completed exercise sessions as required out of the total of 36 sessions (12 weeks * 3 sessions/week). For participants in the TRE+EX combination group, adherence was calculated separately for TRE and EX using the same criteria as above, and the overall adherence rate was determined by averaging the TRE and EX adherence values.” [Page 16, Ln 573-580]

In the Result part: “Adherence to the intervention protocols was assessed for each group over the 12-week period. In the TRE group, adherence was calculated based on participants’ daily eating windows. On average, participants in the TRE group achieved an average adherence rate of 86.9%. For the EX group, participants completed an average of 83.7% of the prescribed sessions (36 sessions over 12 weeks). In the combination group (TRE+EX), adherence was assessed separately for TRE and EX components. Participants in the TRE+EX group achieved an average adherence rate of 89.3% for TRE and 79.6% for EX, resulting in an overall average adherence rate of 84.6% for this group.” [Page 9, Ln 262-268; Figure 3]

Furthermore, we included non-compliant participants in both intention-to-treat (ITT) and per-protocol (PP) analyses. The ITT analysis included all randomized participants, regardless of adherence levels, thus reflecting the intervention’s real-world applicability. In contrast, the PP analysis included only those participants who completed all baseline and post-intervention assessments. Because we did not establish explicit criteria for defining “compliance” in a PP subset before the study’s inception, we decided not to present a separate analysis exclusively for compliant participants. This approach helps avoid potential biases associated with selective reporting and enhances the transparency and rigor of our findings.

Comment 7

Missing data: How did you handle missing data?

Reply: Thank you for raising this question. In our study, we used Generalized Estimating Equations (GEE) as the primary data analysis method. GEE naturally handles missing data by using all available data from each participant without requiring imputation or removal of participants with incomplete data.

Specifically, GEE accounts for missing data by incorporating all observed values into the estimation of population-level parameters, ensuring that the analysis remains valid even when some data points are missing. This approach minimizes the risk of bias caused by missing data and does not require additional methods to manage it.

We appreciate your comment, which allowed us to clarify this aspect of our methodology.

Reviewer #3 (Remarks to the Author):

Reply: Thank you for sharing this information. We greatly appreciate the time and effort that both you and the co-reviewer have dedicated to providing thoughtful and constructive feedback on our manuscript. Your comments have been highly valuable in helping us improve the quality of our work.

Reviewer #4 (Remarks to the Author):

The authors present an RCT examining the combination of time-restricted eating and exercise vs either separate and vs neither. While the trial procedure and chosen outcomes are reasonable and well-documented, and the findings are interesting, there are problems with the analysis (especially the complete lack of control for multiple comparisons) and gaps in the presentation (in both the text of the manuscript and the analysis plan in the protocol) that need to be addressed.

Reply: Thank you for your thoughtful and detailed feedback on our manuscript. We greatly appreciate your recognition of the trial procedure, chosen outcomes, and interest in our findings. At the same time, we fully acknowledge your concerns regarding the lack of control for multiple comparisons and the gaps in the presentation of the manuscript and protocol. These revisions include a more comprehensive description of the statistical analysis plan, clearly outlining how outcomes were analyzed and reported. We are grateful for your constructive remarks, which have allowed us to improve the quality and transparency of our work.

In Figure 1, were outcomes followed for those who discontinued the intervention? If not, why not? If the goal was to test these regimens in a real-world setting, inability/unwillingness to maintain a regimen is part of that, and analyzing as randomized (intent to treat), including everyone not lost to follow-up, makes the most sense, so if there

is outcome data from those who discontinued the intervention, it would be good to see an as-randomized analysis. Also, whatever decisions were made around participants discontinuing their assigned regimen – whether to continue collecting outcome data and whether to include such participants in analyses – should have been decided prior to unblinding the data and described in the protocol.

Reply: Thank you for your thoughtful and insightful comment. We fully acknowledge that analyzing data as randomized, using an intention-to-treat (ITT) approach, is essential to accurately reflect real-world conditions, particularly when participants discontinue the intervention. We appreciate the opportunity to address this critical point and provide clarity on how these situations were handled in our study.

In total, six participants discontinued the intervention during the study. Unfortunately, these participants were lost to follow-up, and we were unable to collect their post-intervention outcome data. Among these six participants: Four initially provided a reason for discontinuing (e.g., time constraints, dissatisfaction with group allocation) but subsequently did not respond to further contact attempts. Two participants directly ceased communication and were lost to follow-up without providing any explanation. We have provided the detail information in Figure 1.

We recognize that decisions regarding the handling of participants who discontinued the intervention or were lost to follow-up should ideally be pre-defined in the study protocol. While our protocol specified the use of generalized estimating equations (GEE) for data analysis, it did not explicitly outline the inclusion of participants who discontinued the intervention. We acknowledge this as an oversight and sincerely apologize for this lack of detail in the protocol.

To address this: We included all randomized participants regardless of adherence or completion of the intervention in the ITT analysis to maintain the randomized structure of the study, which aligns with the ITT principle and preserve the integrity of the randomized design and minimize potential bias. We documented the reasons for discontinuation and loss to follow-up in the flowchart (Figure 1) and transparently described these decisions in the manuscript. We also performed per-protocol (PP) analysis as a sensitivity analysis that we included only participants who completed all assessments (baseline and post-intervention), regardless of adherence to the assigned intervention. Since adherence thresholds were not pre-specified in the protocol, this approach avoids introducing bias or selective reporting at the revision stage and provides insights into the outcomes under “ideal conditions” (full data availability).

Details as: “All randomized participants were included in the analysis according to Intention-to-treat principle. Missing data were not replaced, as the generalized estimating equations (GEE) model applies a natural and appropriate way to accommodate missing data. Primary outcome, secondary outcomes, and the additional monitor parameters were analyzed by a GEE model using group and time as main effects and baseline values, age, menopausal status, and family history of cardiovascular disease included as covariates. Pairwise comparisons using linear contrast were performed to compare the differences between the

intervention groups (TRE+EX vs. CON, TRE+EX vs. TRE, TRE+EX vs. EX, TRE vs. CON, and EX vs. CON) whenever a group × time interaction effect was detected. To account for multiplicity, the Holm-Bonferroni stepwise procedure was applied.” & “Additionally, a per-protocol sensitivity analysis, restricted to participants with complete data across all assessments, was conducted to compare primary, secondary, and monitor parameters among groups.” Page 16, Ln 583-592 & 594-596

We hope this explanation addresses your concerns and demonstrates our commitment to transparency, rigor, and accurately reflecting real-world conditions in our study. Thank you again for raising these important points, which have helped us strengthen the quality and clarity of our work.

The protocol notes that assessors were blinded but makes no mention of others. Did anyone involved in operational decisions have access to unblinded data while the trial was underway?

Reply: Thank you for raising this important and insightful question. We confirm that all operational decisions during the trial were made by one of our co-authors, who remained fully blinded to the entire dataset throughout the trial. Strict measures were in place to ensure that no unblinded data were accessed by any individuals involved in operational decision-making at any stage of the trial. This blinding procedure was rigorously maintained to uphold the integrity of the study and minimize any risk of bias.

We appreciate the opportunity to clarify this aspect of our protocol and hope this explanation addresses your concern.

According to the analysis plan in the protocol, each outcome was examined across 5 pairwise between-group comparisons, but no adjustments for multiple comparisons were made, so the Type 1 error control is inadequate. In a clinical trial, Type 1 error is normally controlled at a family-wise error rate (FWER) of 5% across all comparisons involving the primary outcome(s). Using a simple Bonferroni correction, the significance threshold for comparison in the primary outcome should have been $.05/5 = .01$. The conclusions still appear to hold, but the treatment differences should be presented with 99% confidence intervals ($99\% = 1 - .01$).

Reply: Thank you for your valuable comment regarding the need to adjust for multiple comparisons in our analysis. We appreciate your reminder that controlling the family-wise error rate (FWER) is critical in clinical trials to minimize the risk of Type 1 error, particularly across multiple pairwise comparisons of the primary outcome(s).

To address this concern, we have now applied the Holm-Bonferroni stepwise procedure to adjust for multiple comparisons. This method ensures appropriate control of the FWER while maintaining statistical rigor. We recalculated the significance thresholds for the five pairwise between-group comparisons, applying the Holm-Bonferroni procedure as follows:

For the smallest p-value, the threshold was $0.05/5 = 0.01$.

For the second smallest, $0.05/4 = 0.0125$.

For the third smallest, $0.05/3 = 0.0167$.

For the fourth smallest, $0.05/2 = 0.025$.

For the largest, 0.05.

The p-values for the five pairwise comparisons were ranked in an ascending order as follows: <0.001 , <0.001 , <0.001 , 0.001, and 0.038. All p-values meet their respective adjusted significance thresholds under the Holm-Bonferroni procedure, confirming that the conclusions of our study remain robust and statistically significant even after adjustment for multiple comparisons.

In addition, we have updated the manuscript to reflect these adjustments. Specifically, the Results sections and the relevant tables have been revised, and treatment differences are now reported with 99% confidence intervals (corresponding to the revised significance threshold of 0.01 for the primary outcome). This change enhances the transparency and interpretability of our findings, as recommended.

We are sincerely grateful for your feedback, which has strengthened the rigor and clarity of our analysis. Thank you for your thoughtful suggestions, which have significantly improved the quality of this manuscript.

The secondary outcomes are more problematic, as there are many of them and each one involves 5 comparisons. I suspect a purist might demand a 5% FWER across the entire set of secondary outcomes, but realistically, controlling at 5% separately among each category of outcomes seems reasonable. For other body composition parameters, there are 5 outcomes leading to 25 comparisons; a Bonferroni correction would put the significance threshold at $.05/25 = .002$, which would eliminate some differences in Table 2. (The authors could also consider a less conservative approach that still adequately controls the FWER such as the Hochberg step-up procedure.) Similarly, there are 25 comparisons among the 5 glycemic control outcomes, 50 comparisons among the 10 cardiometabolic markers, and 15 comparisons among the 3 questionnaires, so the necessary corrections will remove a number of significant findings across these outcomes. (The associated confidence intervals should also change; though looser than the proper significance thresholds, I'd be fine with 99% CIs since readers will find them more familiar and understandable than CIs like 99.8%, 99.9%, and 99.67%.)

Reply: We sincerely thank the reviewer for providing such a thoughtful and detailed suggestion regarding the handling of multiple comparisons across the secondary outcomes. This is indeed a critical consideration, and we greatly appreciate the opportunity to enhance the rigor and clarity of our analysis.

To address the issue of multiple comparisons comprehensively, we have implemented the Holm-Bonferroni stepwise procedure to adjust for multiple comparisons across all categories of secondary outcomes, including body composition parameters, glycemic control outcomes, cardiometabolic markers, and questionnaire results. This approach ensures robust control of the family-wise error rate (FWER) while maintaining statistical

rigor. Consequently, we have updated the **Results section** and revised the **relevant tables** in the manuscript to present the adjusted findings based on this procedure.

Furthermore, in alignment with the reviewer's insightful recommendation, we have provided 99% confidence intervals throughout the manuscript. We agree that these intervals strike a balance between statistical correctness and interpretability for readers, as they are more familiar and comprehensible than confidence intervals with unconventional thresholds (e.g., 99.8%, 99.9%, or 99.67%). These updates have been carefully incorporated to enhance the transparency and robustness of our findings.

Once again, we are deeply grateful for the reviewer's constructive feedback, which has significantly contributed to improving the quality and clarity of our manuscript. We hope that these revisions adequately address your concerns and demonstrate our commitment to producing a rigorous and well-communicated analysis.

Pre/post differences are presented with inferential tests in the tables, but those tests are not pre-specified in the protocol. At the very least, the significance tests would need to be adjusted for having 4 comparisons within each outcome; I leave the question of whether it is appropriate to include inferential analyses in this manuscript that were not pre-specified in the trial protocol to the editors.

Reply: Thank you for your thoughtful comment regarding the pre/post differences and the inferential tests presented in the tables. We acknowledge your concern that these analyses were not pre-specified in our original protocol, and we agree that multiple comparisons should be accounted for when evaluating pre/post group differences across four intervention groups.

As outlined in our trial protocol, we used generalized estimating equations (GEE) for data analysis, which included the evaluation of main effects (group and time) and their interaction. When a significant group-by-time interaction effect was identified, we performed post-hoc comparisons using inferential tests to better understand the nature of these interaction effects. At the time of manuscript preparation, we decided to report the inferential results of these post-hoc comparisons to make the findings clearer and easier for readers to interpret, rather than presenting all the group-by-time interaction results. However, we acknowledge that this approach may have deviated from the transparency expected in pre-specified analyses.

In response to your comment, we have decided to revise the manuscript to include a more explicit focus on group-by-time interaction effects as the primary analysis. Instead of examining post-intervention group differences in isolation, we have now presented these interaction effects clearly in the main text and tables. When the group-by-time interaction was significant, we performed post-hoc analyses to identify specific group differences at each time point. Given that there are four intervention groups, we recognized the need to control for multiple comparisons, we provided appropriate adjustments (Holm-Bonferroni stepwise procedure) for multiplicity to control for Type 1 error arising from the multiple comparisons.

These revisions will ensure that our analyses are presented with greater transparency and rigor while maintaining clarity for the readers. We hope this approach sufficiently addresses your concerns, but we remain open to further feedback or suggestions from you or the editors. Thank you again for your thoughtful and constructive comments.

“Energy intake and macronutrient composition” and “Eating window change” are in the results under secondary outcomes, but only the former is described in the methods, and neither appears in Table 3 or in the protocol. The main findings paper for a clinical trial normally only includes pre-specified outcomes and analyses notes in the protocol or a statistical analysis plan.

Reply: Thank you for your insightful feedback and for highlighting the need for greater clarity regarding the inclusion of “Energy intake and macronutrient composition” and “Eating window change” in the Results section. We appreciate the opportunity to address your concerns and provide further clarification.

We acknowledge that “Eating window change” was not pre-specified outcomes in the protocol or statistical analysis plan. Instead, these variables were additional monitor parameters collected as part of our study to better understand the real-world context in which the time-restricted eating (TRE) intervention was implemented.

Energy intake and macronutrient composition were described in the Methods section as part of our monitor parameters. Although these variables were not pre-specified as outcomes, they provide valuable insights into participants’ dietary patterns and the potential effects of the intervention. Therefore, we conducted the same data analysis for these parameters to better understand the dietary changes associated with the intervention. Similarly, eating window change was not pre-specified as an outcome in the protocol or the Methods section, but it served as a key behavioral monitoring variable in our study. Given the central focus of TRE, we tracked participants’ eating windows daily to assess intervention adherence and to capture any behavioral changes related to eating windows. While not initially intended as an outcome, reporting these data helps contextualize participants’ adherence to TRE and provides meaningful insights into the feasibility and real-world implementation of this dietary approach in a free-living setting.

Additionally, in response to another reviewer’s request, we have now included data on moderate-to-vigorous physical activity (MVPA), which was also a monitor parameter in our study. Like the dietary data, MVPA was not pre-specified as an outcome, but its inclusion provides a more comprehensive understanding of participants’ behaviors during the intervention.

To improve transparency and clarity, we have revised the manuscript to explicitly identify these parameters as monitoring measures rather than pre-specified outcomes as: “We reclassified energy intake and macronutrient distribution as additional monitor parameters to better reflect their role in providing supplementary insights rather than serving as key endpoints. To enhance measurement feasibility and reflect the free-living nature of our intervention, we also incorporated two new monitor parameters: eating window and

moderate-to-vigorous physical activity (MVPA) level. This change does not affect the study's primary or secondary outcome analyses or conclusions.” Page 15, Ln 543-548

We have also included the relevant data in the appropriate table to ensure all results are clearly presented. While these parameters fall outside the scope of pre-specified outcomes, we believe their inclusion enhances the ecological validity of our study and provides valuable insights into the real-world implementation of TRE. We are grateful for your thoughtful comments, which have helped us refine the manuscript and improve the clarity and rigor of our reporting. Thank you for your careful review and for prompting us to strengthen the transparency of our study.

It would be helpful to provide the MCID (minimal clinically important difference) for all the measures where one is known to help the reader decide if the statistically significant differences are also clinically meaningful.

Reply: Thank you for your thoughtful comment regarding the inclusion of the minimal clinically important difference (MCID) for the measures in our study. We appreciate your suggestion, as it highlights the importance of distinguishing between statistical significance and clinical relevance.

For our primary outcome, fat mass, we were unable to identify an established MCID in the literature. However, to provide additional context for clinical interpretation, we included data on body weight change—a related measure for which the MCID is more commonly defined. As noted in the Discussion section, the MCID for weight loss is typically considered to be a 5% reduction in body weight. While our intervention achieved a mean weight loss of 4.4%, falling slightly below this threshold, we have clearly acknowledged this in the Discussion to help readers interpret the clinical significance of our findings.

Additionally, in Figure 2, we provide the change values for both fat mass and body weight. To further enhance the clarity of our findings, we have included a reference line in the body weight change graph to indicate the 5% threshold, enabling readers to directly compare our results to this clinically significant benchmark.

For the other outcomes assessed in our study, while we observed statistically significant differences for several measures, we were unable to identify established MCIDs for these specific outcomes in the literature. Despite the absence of MCIDs, we believe that reporting these significant findings contributes valuable insights into the potential effects of our intervention, and we encourage further research to better define clinically meaningful thresholds for these measures.

We are grateful for your suggestion, which has encouraged us to further emphasize the distinction between statistical and clinical significance in our manuscript. Thank you for helping us enhance the clarity and interpretability of our findings.

The captions for Figures 2 and 3 should provide more information. Are the error bars confidence intervals (and if so, what width), standard deviations, standard errors, or something else? Also, all abbreviations used should be explained in the legend for each figure they appear in even if they're also defined in the text. For Figure 3, plotting the group

differences and their confidence intervals would probably be more useful than plotting the group means (especially since the pre, post, and change scores are presented in tables for the other outcomes). But as a more general question, why are these two secondary/exploratory (it's unclear as noted above) outcomes given figures while the primary and main secondary outcomes are not? That seems odd, since a reader would assume the figures show what the authors consider the most important findings of the paper.

Reply: Thank you for your thoughtful comments regarding Figures 2 and 3. We appreciate your detailed feedback and the opportunity to improve the clarity and focus of our figures. We recognize that the inclusion of Figures 2 and 3 in their original form may have drawn attention away from our primary outcomes, which we consider the most important findings of our study. In response to your comment, we have added a new figure that highlights the primary findings related to fat mass and body mass, which are central to the objectives of our study. This new figure (Update Figure 2) ensures that the most critical results are visually represented for the reader.

Regarding the original Figures 2 and 3, we acknowledge that the error bars lacked sufficient detail in the figure legends. We have revised these figures and their captions to clearly explain what the error bars represent (e.g., confidence intervals, standard errors, etc.). Additionally, all abbreviations used in the figures are now fully defined in the legends to ensure clarity, even if they are explained elsewhere in the text.

To further refine the presentation of our findings, we have updated the figures that provide deeper insights into the behavioral changes and real-world feasibility of the intervention. Specifically, we now include figures (Update figure 3) showing the daily energy intake, eating window change, moderate-to-vigorous physical activity (MVPA) change, and adherence rate in TRE, EX, and separate/combine in the TRE+EX group. These updates reflect the nature of our study's free-living context and provide readers with a more comprehensive understanding of participants' behavioral changes and the intervention's feasibility.

We believe these updates provide a clearer and more balanced presentation of the study findings. Our revised figures now emphasize the effectiveness of the intervention for the primary outcomes while also illustrating the feasibility and behavioral changes observed in this free-living context. Thank you again for your valuable feedback, which has helped us enhance the focus and clarity of our manuscript.

I believe these issues can all be addressed, and it looks like your primary findings (though not all of the secondaries) will survive the necessary adjustments for multiple comparisons. I look forward to seeing the next iteration.

Reply: Thank you for your thoughtful feedback and for highlighting the areas that require further attention. Your input has been invaluable in guiding us toward improving the rigor and clarity of our work. We look forward to sharing the revised manuscript with you.

REVIEWERS' COMMENTS

Reviewer #1 (Remarks to the Author):

Thank you for your revision. I believe they have strengthened the paper. There are still a couple of issues that need to be resolved.

Reply: Thank you for your kind feedback and for noting the improvements in our revised paper. We appreciate your comments on the remaining issues, and we have carefully considered all comments to improve the clarity of our manuscript. Below, please find the detailed responses.

Major:

1. Thank you for clarifying how the TRE intervention was done. However, the flexible TRE window needs to be mentioned throughout the paper as allowing a flexible window contradicts the primary component of TRE. TRE requires a consistent eating window as defined by the International Consensus of Fasting ([https://www.cell.com/cell-metabolism/fulltext/S1550-4131\(24\)00269-9](https://www.cell.com/cell-metabolism/fulltext/S1550-4131(24)00269-9)). To be more accurate, it should be referred to as 16:8 intermittent fasting which allows the eating window to move. However, if the authors insist on calling it TRE, then in the title, abstract, intro, and discussion, it needs to be made clear that this is a modified form of TRE. Examples include calling it flexible TRE (flexTRE) or modified TRE (mTRE), but it should not be referred to as TRE alone as it is very misleading.

Reply: We thank the reviewer for this crucial observation regarding the precise definition of TRE. We agree that our intervention, which allows for a shifting eating window, represents a modification of the traditional TRE protocol and that using the term 'TRE' alone could be misleading. To ensure our manuscript is accurate and clear, we have adopted the reviewer's suggestion. We have systematically revised the term to 'flexible TRE (flexTRE)' throughout the entire manuscript, including the title, abstract, introduction, methods, and discussion. We believe this change more accurately describes our study design and strengthens the paper.

Minor:

1. In the abstract and introduction, you should include the duration of the TRE interval (ex: 8 hour TRE) and when they were allowed to eat (set time or personalized)

Reply: We thank the reviewer for their constructive feedback. We agree that specifying the duration and flexibility of the TRE protocol in the abstract and introduction is crucial for clarity. Accordingly, we have revised the manuscript to include these important details.

In abstract, we have revised the text as: "This study investigates the effects of 8-hour flexible time-restricted eating (flexTRE)..." [Page 1, Ln 16-17]

In the introduction, we now state: “To address this, the present study implements a flexible TRE (flexTRE) protocol, allowing participants to self-adjust the timing of their 8-hour eating window daily to accommodate the variability of everyday life.” [Page 3, Ln 114-116]

2. Figure 1 quality needs to be improved. This may just be because it is in the manuscript.

Reply: We thank the reviewer for pointing out the quality issue with Figure 1. To resolve this, we have uploaded a new, high-resolution version of Figure 1 as a separate file.

3. Figure 2: the axis should have the same scale for a given outcome. Example: the axis scale for all fat mass changes should be the same and all body mass change should be the same so the reader can easily see the differences

Reply: We are grateful to the reviewer for this valuable feedback on data visualization. We have revised Figure 2 in response. As suggested, we have standardized the axis scales for each outcome across all groups. We agree that this significantly improves the figure's readability and makes the comparisons much more direct for the reader. Please see the revised Figure 2.

4. It's interesting that fat-free mass only decreased in the TRE+EX group and not in TRE or EX alone. How do you explain this? It would be good to add this to the discussion.

Reply: We thank the reviewer for highlighting this important and interesting finding. We agree that this observation warrants a thorough explanation. As suggested, we have added a new paragraph to the Discussion section to address this point.

The following text has been added to the Discussion: “We observed a modest (<0.5 kg) but statistically significant reduction in FFM across all three intervention groups before correcting for multiple comparisons. This reduction was proportional to total weight loss, which is consistent with the established physiological observation that approximately 25% of lost weight is derived from FFM, suggesting the finding may not be clinically meaningful. After applying the Holm procedure for multiple comparisons, only the combined flexTRE+EX group retained a statistically significant reduction in FFM. The loss of significance in the flexTRE and EX groups alone likely reflects limited statistical power to detect small effects in individual interventions. Future randomized controlled trials with larger sample sizes and longer follow-up durations are warranted to further evaluate the complex interplay between these interventions and FFM.” [Page 8, Ln 312-321]

We believe this detailed explanation provides a scientifically sound interpretation of the data, and we thank the reviewer again for prompting this valuable addition to the manuscript.

Reviewer #2 (Remarks to the Author):

I thank the authors for thoroughly responding to my review report. I do not have further concerns or comments.

Reply: We are grateful for the positive feedback from Reviewer #2. We are pleased that our revisions successfully addressed the initial concerns that were raised. The contribution from Reviewer #2 has been valuable in improving our manuscript, and we offer our sincere thanks.

Reviewer #3 (Remarks to the Author):

Reply: We thank Reviewer #3 for this clarification. We sincerely appreciate the contribution of the co-reviewing process.

Reviewer #5 (Remarks to the Author):

Thank you for the opportunity to review the revised version of the manuscript.

I have carefully examined the authors' responses and the updated manuscript. I am pleased to note that all previous comments of reviewer 4 have been adequately addressed. The revised version has improved the clarity and quality of the work, and I only have two minor comments:

Reply: Thank you for the kind feedback and for noting the previous comments of reviewer 4 have been addressed. We appreciate your comments on the remaining issues, and we have carefully considered all comments to improve the clarity and quality of our manuscript.

1. The abbreviation "CON" in the abstract appears without an initial definition. For clarity, please provide the full term at its first mention, followed by the abbreviation in parentheses, e.g. control (CON).

Reply: We thank the reviewer for this observation. We have revised the abstract to define the control group abbreviation as suggested: "A total of 104 participants were randomized to flexTRE, EX, flexTRE+EX, and control group (CON) in a 1:1:1:1 ratio." [Page 1, Ln 18-20]

2. To assist readers who may wish to explore the technical details, please consider citing the following reference as the original source introducing the Holm-Bonferroni method.

Holm, S. (1979). "A simple sequentially rejective multiple test procedure". *Scandinavian Journal of Statistics*. 6 (2): 65–70.

Reply: We thank the reviewer for this valuable suggestion to cite the original source for the Holm-Bonferroni method. We agree this is an important addition for methodological transparency. We have revised the data analysis subsection to include the citation. The full reference has also been added to our bibliography.